# The kinesin-3 KIF1C undergoes liquid-liquid phase separation for accumulation of specific transcripts at the cell periphery

Qi Geng(耿奇) [ID][1], Jakia Jannat Keya [ID][2], Takashi Hotta[2] & Kristen J Verhey [ID][2✉]

## Abstract

In cells, mRNAs are transported to and positioned at subcellular areas to locally regulate protein production. Recent studies have identified the kinesin-3 family member motor protein KIF1C as an RNA transporter. However, it is not clear how KIF1C interacts with RNA molecules. Here, we show that the KIF1C C-terminal tail domain contains an intrinsically disordered region (IDR) that drives liquid–liquid phase separation (LLPS). KIF1C forms dynamic puncta in cells that display physical properties of liquid condensates and incorporate RNA molecules in a sequence-selective manner. Endogenous KIF1C forms condensates in cellular protrusions, where mRNAs are enriched in an IDR-dependent manner. Purified KIF1C tail constructs undergo LLPS in vitro at near-endogenous nM concentrations and in the absence of crowding agents and can directly recruit RNA molecules. Overall, our work uncovers an intrinsic correlation between the LLPS activity of KIF1C and its role in mRNA positioning. In addition, the LLPS activity of KIF1C's tail represents a new mode of motor-cargo interaction that extends our current understanding of cytoskeletal motor proteins.

**Keywords** Microtubule; Kinesin; LLPS; mRNA Transport; IDR
**Subject Categories** Cell Adhesion, Polarity & Cytoskeleton; Membranes & Trafficking; RNA Biology

## Introduction

Subcellular targeting of mRNAs is prevalent and can be used to spatially and temporally regulate the processing, translation, and stability of mRNA in cells (Ryder and Lerit, 2018; Engel et al, 2020; Lashkevich and Dmitriev, 2021; Das et al, 2021; Gasparski et al, 2022). For example, β-actin mRNA is distributed to the leading edge of migrating cells where local translation can produce abundant actin required for lamellipodium formation (Jeffery et al, 1983; Lawrence and Singer, 1986; Kislauskis et al, 1997; Hüttelmaier et al, 2005; Katz et al, 2012; Liao et al, 2015). Subsequent studies found numerous mRNAs that are enriched in protrusions of migratory cells (Chouaib et al, 2020; Mili et al, 2008;

Mardakheh et al, 2015), such as the small GTPase Rab13 whose mRNA distribution affects its co-translational association with different protein complexes during cell migration and tissue morphogenesis (Ioannou and McPherson, 2016; Moissoglu et al, 2020; Costa et al, 2020). Subcellular targeting of mRNAs has also been shown to be critical in differentiated cells such as oligodendrocytes and neurons (Perez et al, 2021; Li et al, 2021; Jung et al, 2023; Gala et al, 2023). Subcellular targeting of mRNAs has several advantages over localization of their protein products, such as the ability to regulate translation in response to local stimuli, and to prevent proteins from undesired interactions or premature functions before reaching their destination.

In general, there are two mechanisms for subcellular targeting or compartmentalization of mRNAs. First, mRNAs can be enriched into membrane-less bodies called RNA granules through liquid–liquid phase separation (LLPS) (Tian et al, 2020; Rhine et al, 2020; Roden and Gladfelter, 2021). LLPS is the process of molecules working against entropy to cluster into a dense phase termed a biomolecular condensate that is separated from the surrounding diffusive phase. LLPS is driven by weak, multivalent interactions between molecules that typically contain intrinsically disordered regions with low-complexity sequences (Hyman et al, 2014; Shin and Brangwynne, 2017). Their highly dynamic nature and sensitivity to the chemical environment empowers LLPS condensates to regulate translational activity as well as the degradation of mRNAs (Hyman et al, 2014; Shin and Brangwynne, 2017; Wiedner and Giudice, 2021; Parker et al, 2022).

Second, mRNAs can be targeted via active transport by molecular motor proteins (Goldman and Gonsalvez, 2017; Sahoo et al, 2018; Suter, 2018; Turner-Bridger et al, 2020; Rodrigues et al, 2021). For microtubule-mediated long-distance transport, kinesin family (KIF) proteins distribute cargoes towards microtubule plus ends in the cell periphery, while dynein proteins drive transport in the opposite direction. mRNAs are thought to form complexes with molecular motors via RNA-binding adapter proteins such as Egalitarian (Dienstbier et al, 2009). During transport, mRNAs can present as single molecules or as clusters in granules (Liao et al, 2019; Tübing et al, 2010; Barbarese et al, 1995; Kanai et al, 2004; Mikl et al, 2011; Batish et al, 2012). mRNAs can also be transported by hitchhiking on organelles such as endosomes and lysosomes (Baumann et al, 2012; Liao et al, 2019; Corradi et al, 2020). How motor proteins bind to and transport specific mRNAs and other RNA species remains an important question in the field.

[1]Department of Molecular, Cellular, and Developmental Biology, University of Michigan, Ann Arbor, MI, USA. [2]Department of Cell and Developmental Biology, University of Michigan Medical School, Ann Arbor, MI, USA. ✉E-mail: kjverhey@umich.edu

Recent work has shown that KIF1C, a ubiquitously expressed member of the kinesin-3 family, functions in mRNA transport in numerous cell types (Pichon et al, 2021; Nagel et al, 2022; Norris and Mendell, 2023; Hildebrandt et al, 2023). KIF1C had previously been implicated in the transport of Golgi-derived vesicles, integrins, and secretory vesicles (Dorner et al, 1998; Schlager et al, 2010; Theisen et al, 2012), and thus appears to be a kinesin protein capable of transporting both membrane-bound organelles and membrane-less complexes.

During cell migration, specific mRNAs are localized to protrusions at the leading edge (Mingle et al, 2005; Mili et al, 2008). While the mRNAs of most motor proteins localize randomly throughout the cytoplasm, the mRNAs of three kinesins were found to localize specifically to protrusions: the kinesin-1 *KIF5B*, the kinesin-3 *KIF1C*, and the kinesin-4 *KIF4A* (Chouaib et al, 2020). Interestingly, KIF1C was the only kinesin where the protein and mRNA co-localized at the protrusion (Chouaib et al, 2020). Immunoprecipitation of KIF1C identified a number of co-precipitated mRNAs including the protrusion-localized mRNAs of *RAB13*, *NET1*, *TRAK2*, and *KIF1C* itself (Pichon et al, 2021). Two-color imaging showed that KIF1C directly transports mRNAs and drives the formation of peripheral, multimeric RNA clusters (Pichon et al, 2021). Recently, KIF1C was found to interact with several components of the exon junction complex (EJC), which binds to mRNAs throughout their life cycle, and loss of KIF1C activity resulted in decreased transport of EJC components along neuronal processes (Nagel et al, 2022). KIF1C was also found to interact with Muscleblind-like (MBNL) proteins involved in splicing, polyadenylation, and localization of mRNAs in numerous cell types (Hildebrandt et al, 2023). Whether KIF1C binds to adapter proteins to recruit specific mRNAs or binds to specific mRNAs that are associated with adapter proteins is largely unknown. Support for the former possibility comes from recent finding that KIF1C's association with MBNL occurs in an RNA-independent manner (Hildebrandt et al, 2023). However, several studies support the latter possibility as mRNA-protein cross-linking approaches identified KIF1C as a direct binder of mRNAs (Nagel et al, 2022; Baltz et al, 2012; Castello et al, 2012) and binding between KIF1C and EJC components is RNA-mediated (Nagel et al, 2022).

In this study, we show that the C-terminal tail domain of KIF1C is an intrinsically disordered region (IDR) with low-complexity sequence and a putative prion-like domain (PLD). The IDR is necessary and sufficient for LLPS and the formation of KIF1C condensates with liquid or hydrogel properties. To our knowledge, this is the first example of a kinesin undergoing LLPS via its IDR. Synthesized RNA oligos and endogenous *RAB13* and *NET1* mRNAs can be enriched into KIF1C condensates in a sequence-selective manner. Overall, our findings provide a novel mechanism of motor-RNA interaction through LLPS, highlighting the behavior of the disordered KIF1C C-terminal tail domain that is unique among the kinesin superfamily.

## Results

### KIF1C forms dynamic puncta via its C-terminal intrinsically disordered tail domain

To probe the localization of KIF1C in mammalian cells, we expressed fluorescently-tagged full-length KIF1C (Fig. 1A) in COS-7 cells by transient transfection. KIF1C localized diffusely throughout the cell as

well as to punctate structures (Fig. 1B). Other fluorescently-tagged kinesin-3 family members also localized diffusely and to organellar structures, for example, KIF1Bβ to lysosomes (Matsushita et al, 2004), KIF13B to Rab6-positive secretory vesicles (Serra-Marques et al, 2020), and KIF16B to endosomes (Hoepfner et al, 2005) (Fig. 1B; Appendix Fig. S1A). In contrast to the other kinesin-3s, the KIF1C puncta appeared to be heterogenous in size. Indeed, quantification showed that the KIF1C puncta ranged in size from 0.30 to 6.13 μm in diameter (average ± SD: 1.31 ± 0.91 μm, Fig. 1C) whereas KIF16B puncta had a more uniform size distribution (ranging from 0.35 to 1.48 μm in diameter, average ± SD: 0.75 ± 0.17 μm, Fig. 1C). We wondered whether the larger puncta could be formed by aggregation or fusion of smaller puncta. By live-cell imaging, KIF1C puncta were observed to undergo fission into smaller puncta which could then undergo fusion events to generate larger puncta over a time scale of several seconds (Fig. 1D; Movie EV1). This suggests that the KIF1C puncta are dynamic in nature.

To probe the identity of the KIF1C puncta, we stained cells for various markers but found that the KIF1C puncta did not colocalize with markers of endosomes, lysosomes, focal adhesions, or mitochondria (Appendix Fig. S1B). We next examined the protein sequence of KIF1C. Like other kinesins involved in cargo transport, full-length (FL) KIF1C contains an N-terminal motor domain (MD), a stalk domain with several coiled-coil (CC) segments, and a unique C-terminal tail domain for binding specific cargoes (Fig. 1A). KIF1C's tail domain was previously referred to as a proline-rich domain (Siddiqui et al, 2019; Kendrick et al, 2019). Analysis of its amino acid composition reveals low sequence complexity, with significant enrichment not only of proline (20.2%) residues, but also arginine (13.3%), glycine (9.2%), and serine (8.7%) residues (Appendix Fig. S2A,B). Structure prediction using IUPred suggests that the tail domain is an intrinsically disordered region (IDR) and PLAAC analysis indicates that a subregion of the IDR is a prion-like domain (PLD) (Fig. 2A), a type of low-complexity region frequently found in RNA-binding proteins (RBPs) (Mészáros et al, 2018; Lancaster et al, 2014). Notably, with the exception of kinesin-3 KIF1Bα, long stretches of IDR with PLD are not found in the tail domains in other kinesin-3 motors or of kinesin-1 motors (Appendix Fig. S2C,D).

To test whether the IDR is required for the formation of the KIF1C puncta, we generated KIF1C variants with serial truncations from the N-terminus (Fig. 2B) or the C-terminus (Appendix Fig. S3A). All constructs were tagged by mNeonGreen (mNG) and their localization was examined upon transient expression in cultured cells. Deletion of the IDR from the C-terminus of KIF1C (ΔIDR) resulted in a largely diffuse localization (Appendix Fig. S3B). Although this construct did show small accumulations at the cell periphery, these appeared to be filament-shaped accumulations along microtubules, similar to a constitutively-active KIF1C construct containing only the motor domain (MD) (Appendix Fig. S3C). Further C-terminal truncations also failed to form puncta in cells (Appendix Fig. S3B), indicating that the IDR is necessary for puncta formation.

Deletion of the N-terminal motor domain resulted in a stalk+tail (ST) construct (aa 349–1103) that forms puncta localized in the perinuclear region (Fig. 2C). Thus, the motor domain is not required for the formation of KIF1C puncta but drives their transport to the cell periphery. Deletion of increasing lengths of the stalk domain had little to no effect on the formation of the puncta or on their morphology or distribution within the cytoplasm (Fig. 2C). Indeed, the IDR in the tail

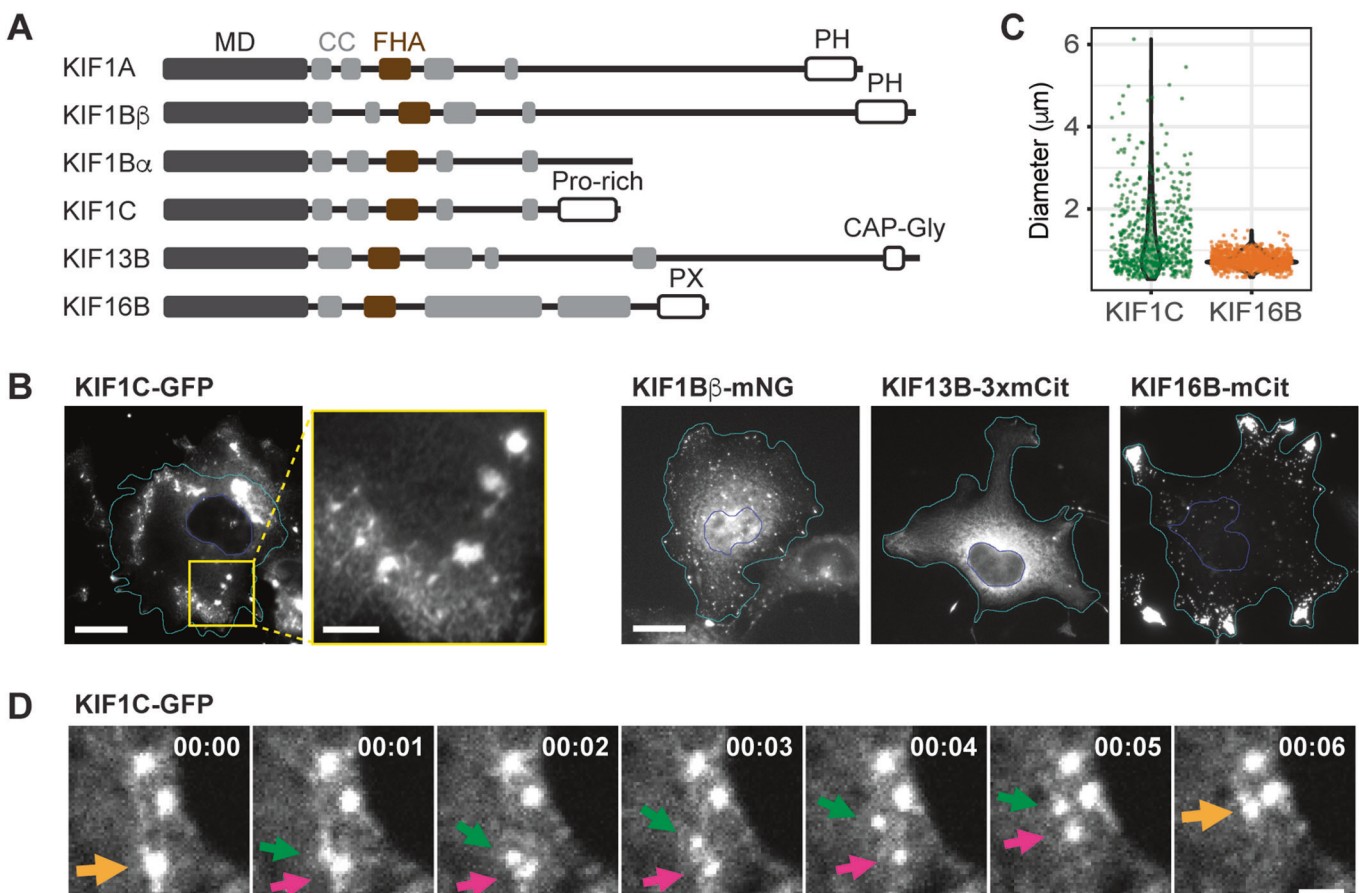

**Figure 1.  KIF1C forms dynamic puncta in cells.**

(A) Schematic of domain organization of the indicated kinesin-3 family members. MD: Motor domain; CC: Coiled-coil domain; FHA: Forkhead-associated domain. PH: Pleckstrin homology domain; Pro-rich: Proline-rich domain; CAP-Gly: Glycine-rich domain of Cytoskeleton-associated proteins; PX: PhoX homology domain. (B) Localization of fluorescently-tagged kinesin-3 motors in COS-7 cells. Scale bar: 20 μm for whole cell views, 5 μm for magnified image of the KIF1C-expressing cell. Cyan lines indicate cell boundaries. Blue lines indicate nuclear boundaries. (C) Quantification of the size distribution of individual KIF1C or KIF16B puncta. KIF1C puncta diameter 1.31 ± 0.91 μm (mean ± SD). KIF16B puncta diameter 0.75 ± 0.17 μm. Each dot represents one punctum. $N = 38$ cells for KIF1C (601 puncta); $N = 34$ cells for KIF16B (1120 puncta). (D) Representative live-cell imaging (see Movie EV1) showing dynamic fusion and fission events of KIF1C puncta. The arrows point out a punctum (yellow) transiently splitting into two puncta (green and magenta) and subsequently merging back into one punctum (yellow). Scale bar: 2 μm. Time label is [min:sec]. Source data are available online for this figure.

domain is sufficient for puncta formation as a construct containing only the IDR (aa 886–1103) formed puncta in the cytoplasm (Fig. 2C). The lower fold enrichment of IDR in puncta as compared to the other constructs (Fig. 2D,E) is likely due to the high nuclear localization of the IDR construct which results in a lower cytoplasmic concentration. Taken together, these results indicate that the C-terminal IDR is necessary and sufficient for KIF1C puncta formation in the cytoplasm.

## KIF1C puncta show properties of biomolecular condensate in cells

The fact that formation of KIF1C puncta requires an IDR suggests that the puncta may be biomolecular condensates formed through the process of liquid–liquid phase separation (LLPS) (Banani et al, 2017; Hyman et al, 2014; Falahati and Haji-Akbari, 2019). Previous work has shown that LLPS is typically mediated by IDR or low-complexity domains that provide weak, multivalent interactions to allow molecules to stick together against

entropy. Other features of the KIF1C puncta, such as their heterogenous size and their active fusion and fission, support the idea that they may be membrane-less, liquid condensates. We thus tested the hypothesis that KIF1C undergoes IDR-mediated LLPS by probing the physical and chemical properties of KIF1C puncta in cells.

We first used fluorescence recovery after photobleaching (FRAP) to test whether there is active exchange of molecules between KIF1C puncta (the dense phase) and the cytoplasm (the diffusive phase). mCherry (mCh)-tagged full-length (FL), stalk +tail (ST), and IDR constructs (Fig. 3A) were expressed in cells together with freely diffusive GFP as a control. Individual puncta were selected and photobleached at high laser power and the fluorescence recovery in the bleached area was measured. All three KIF1C constructs showed rapid fluorescence recovery with average half-time ($\tau_{1/2}$) of 3.5 s for KIF1C(FL), 17.0 s for KIF1C(ST), and ≤1.0 s for KIF1C(IDR) (Fig. 3B,C; Movies EV2, 3 and 4) indicating that KIF1C molecules can undergo dynamic exchange between the

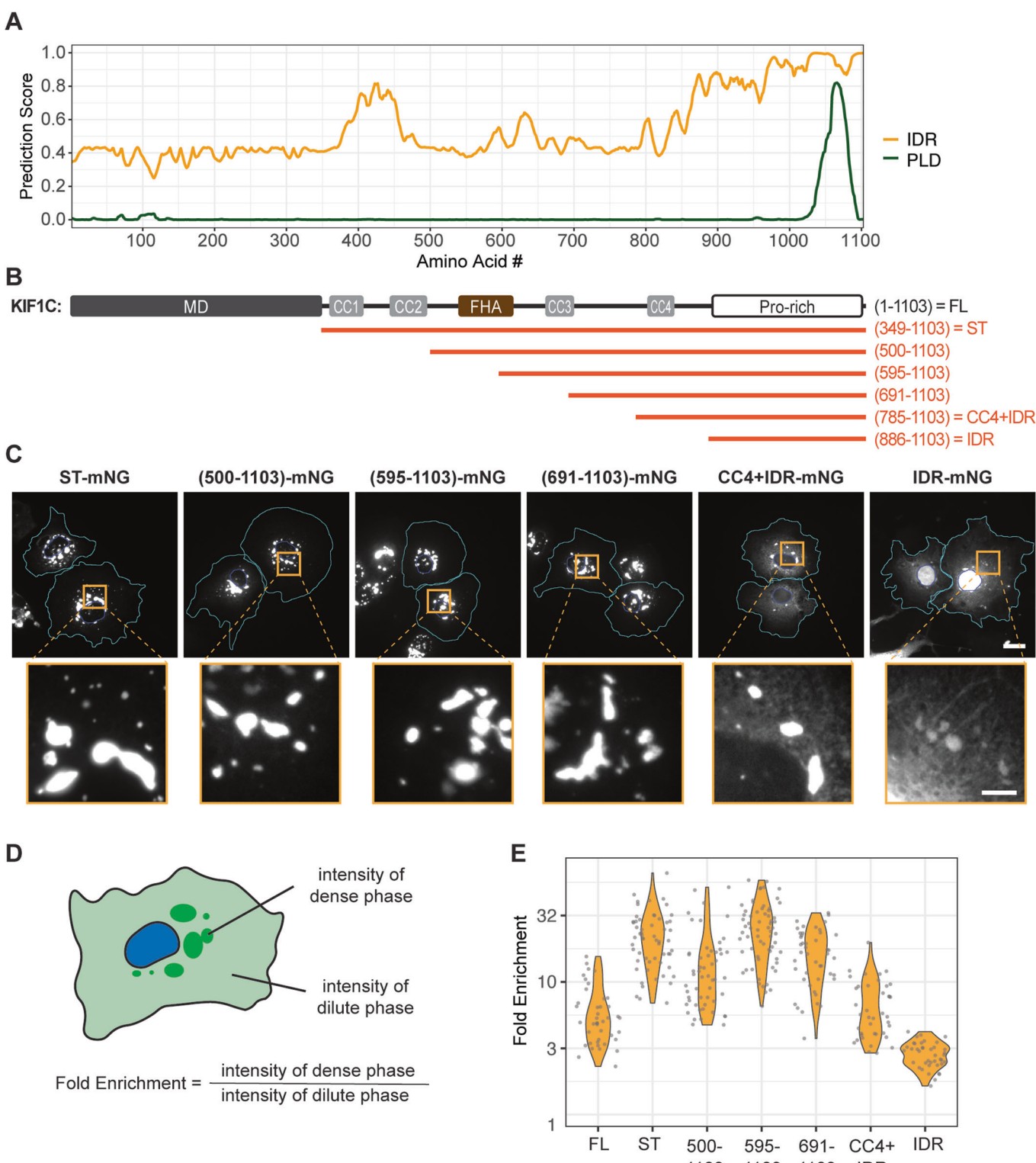

**D** Fold Enrichment = $\dfrac{\text{intensity of dense phase}}{\text{intensity of dilute phase}}$

condensate and the surrounding environment. Note that for KIF1C(IDR) ($\tau_{1/2} \leq 1.0$ s) and GFP ($\tau_{1/2} \leq 0.8$ s), the measured $\tau_{1/2}$ values are likely an overestimation as fluorescence recovery occurred before post-bleaching images could be acquired. There is also an immobile population of molecules in the condensates, particularly for KIF1C(FL) [40% mobile fraction (m.f.)] and

KIF1C(ST) (32% m.f.) (Fig. 3C). These findings suggest that the KIF1C condensates are primarily liquid phase droplets, but can exhibit partial gel-like character, as observed for other biomolecular condensates, including microtubule-associated proteins (King and Petry, 2020; Song et al, 2023; Miesch et al, 2023; Jain et al, 2016; Feric et al, 2016; Shin et al, 2017).

◄ **Figure 2. The C-terminal IDR domain determines KIF1C punctum formation.**

(A) IUPred prediction of intrinsically disordered region (IDR) and PLAAC prediction of prion-like domain (PLD) for KIF1C. x-axis: KIF1C amino acid residue number; y-axis: predicted probability of the given residue being part of an IDR (orange line) or a PLD (green line). (B) Schematic of full-length (FL) KIF1C domain organization, aligned with the x-axis of (A). Orange line segments indicate KIF1C serial truncations from the N-terminus. (C) Localization of KIF1C truncations in COS-7 cells. Scale bar: 20 μm for whole cell views, 5 μm for magnified images. Cyan lines indicate cell boundaries. Blue lines indicate nuclear boundaries. (D) Schematic for quantification of the fold enrichment of KIF1C in puncta. (E) Fold enrichment of full-length or truncated KIF1C. y-axis: fold enrichment plotted on log scale. Fold enrichments (mean ± SD): 6.0 ± 3.2 for FL, N = 42 cells; 22.7 ± 11.1 for ST, N = 58 cells; 14.2 ± 10.5 for (500–1103), N = 54 cells; 26.0 ± 12.7 for (595–1103), N = 65 cells; 16.7 ± 7.5 for (691–1103), N = 45 cells; 6.8 ± 3.6 for CC4 + IDR, N = 40 cells; 2.9 ± 0.6 for IDR, N = 40 cells. Source data are available online for this figure.

We next tested whether the appearance of the KIF1C puncta responds to a change in molecular concentration. To do this, we utilized a cytoplasm dilution assay to rapidly manipulate the cytoplasmic conditions in a reversible manner (Hancock, 2004; Alenquer et al, 2019). In this assay, treatment with hypotonic media induces an increase in cytoplasmic volume and concomitant decrease in cytoplasmic protein concentrations which can be reversed upon switching the media back to isotonic conditions (Fig. 3D). We utilized the stalk+tail construct KIF1C(ST) to prevent motor-driven mixing of KIF1C condensates. Upon application of hypotonic media, the mNG-tagged KIF1C(ST) puncta dissolved within minutes (Fig. 3E,F). Initially, larger KIF1C(ST) puncta disassembled into smaller puncta, followed by dissolution of small puncta into the diffusive phase (Movie EV5). Strikingly, when the solution was returned to isotonic conditions, numerous small puncta appeared in the cytoplasm and then fused into larger puncta that localized in the perinuclear region (Fig. 3E,F). As a control, we tested whether KIF16B puncta undergo reversible dissolution and reformation under these conditions as KIF16B binds directly to membranes via its PX domain (Hoepfner et al, 2005). During the iso-hypo-isotonic cycle, KIF16B puncta did not dissolve or undergo fission/fusion (Appendix Fig. S4A,B, Movie EV6). Overall, we conclude that KIF1C puncta are liquid condensates formed through LLPS.

Recent work has shown that several microtubule-associated proteins can undergo LLPS and "wetting" of the microtubule surface (Tan et al, 2019; Siahaan et al, 2019; Setru et al, 2021; Wu et al, 2021; Miesch et al, 2023; Volkov and Akhmanova, 2023). CLIP-170 is of particular interest as it also undergoes LLPS at microtubule plus ends in the cell periphery (Wu et al, 2021; Miesch et al, 2023). We tested whether the KIF1C condensates are related to those of CLIP-170, however, no colocalization was observed between mCh-tagged KIF1C condensates and GFP-tagged CLIP-170 condensates (Appendix Fig. S4C). Also of interest is the finding that some microtubule-associated proteins can enrich free tubulin in the condensate via their microtubule-binding domains (Jiang et al, 2015; Imasaki et al, 2022; Hernández-Vega et al, 2017; Woodruff et al, 2017; Baumgart et al, 2019). We tested whether KIF1C condensates can enrich tubulin, however, we found that KIF1C condensates do not recruit free tubulin upon depolymerization of cytoplasmic microtubules (Appendix Fig. S4D). Therefore, KIF1C condensates appear to be distinct from other microtubule-associated condensates.

## KIF1C condensates display properties of RNA granules

What does LLPS activity mean for KIF1C's function as a kinesin? Recently, several groups reported that KIF1C associates with and transports mRNAs (Pichon et al, 2021; Nagel et al, 2022; Norris and Mendell, 2023; Hildebrandt et al, 2023). These findings are particularly interesting in light of KIF1C's ability to undergo LLPS, as many phase-separating proteins are known to be RNA-binding proteins (RBPs) (Wiedner and Giudice, 2021; Weber and Brangwynne, 2012). Furthermore, KIF1C contains a putative prion-like domain (PLD, Fig. 2A), a domain that is known to mediate LLPS of RBPs such as FUS and TDP-43 (Carey and Guo, 2022; Patel et al, 2015; Monahan et al, 2017; Murthy et al, 2019; Conicella et al, 2020; Hallegger et al, 2021; Grese et al, 2021). Therefore, we explored whether KIF1C condensates are related to other RNA granules and whether they contain RNAs.

To test whether KIF1C can interact with other RNA granules in cells, we overexpressed KIF1C in COS-7 cells and found that KIF1C colocalizes with protein markers of two types of RNA granules, stress granules, and P-bodies, by immunofluorescence staining (Appendix Fig. S5A). The IDR is necessary and sufficient for interaction with these RNA granules as the KIF1C(IDR) construct co-localized with both stress granules and P-bodies, whereas the KIF1C(ΔIDR) construct did not colocalize with either membrane-less organelle (Appendix Fig. S5A). As a control, we examined whether the related kinesin-3 KIF1Bα, which also contains an IDR and PLD and forms puncta in cells (Appendix Fig. S5B), colocalizes with stress granules or P-bodies but found that KIF1Bα did not colocalize with either RNA granule (Appendix Fig. S5C).

Previous work has suggested that LLPS of prion-like RBPs, such as FUS, TDP-43, and TAF15, is suppressed by high concentrations of nonspecifically-interacting RNAs, and that highly-structured RNAs act as scaffolds that promote the nucleation of condensates within the high RNA environment of the cytoplasm and/or nucleoplasm (Maharana et al, 2018; Duan et al, 2022). Given KIF1C's PLD, we tested whether its LLPS is also sensitive to the non-specific RNA pool. To do this, we microinjected RNase A to remove the cytoplasmic RNA in a non-specific manner. We again used the KIF1C(ST) construct to prevent motor-driven movement along microtubules and carried out live-cell imaging before and for ~20 min after RNase A microinjection. After injection of RNase A, existing KIF1C(ST) condensates showed no change in morphology or distribution but strikingly, new KIF1C(ST) puncta appeared within minutes (Fig. 4A; Movie EV7). The new condensates were dynamic and often underwent fusion to make larger puncta. In contrast, the injection marker rhodamine-dextran did not alter the KIF1C(ST) condensates (Appendix Fig. S6A; Movie EV8). These results demonstrate that KIF1C's LLPS behavior is sensitive to the endogenous cytoplasmic RNA content, and support the idea that specific RNA molecules can induce KIF1C's phase separation in a system buffered by nonspecifically-interacting RNAs.

To test whether KIF1C condensates can incorporate RNA, fluorescently-labeled synthesized RNA oligos were introduced into

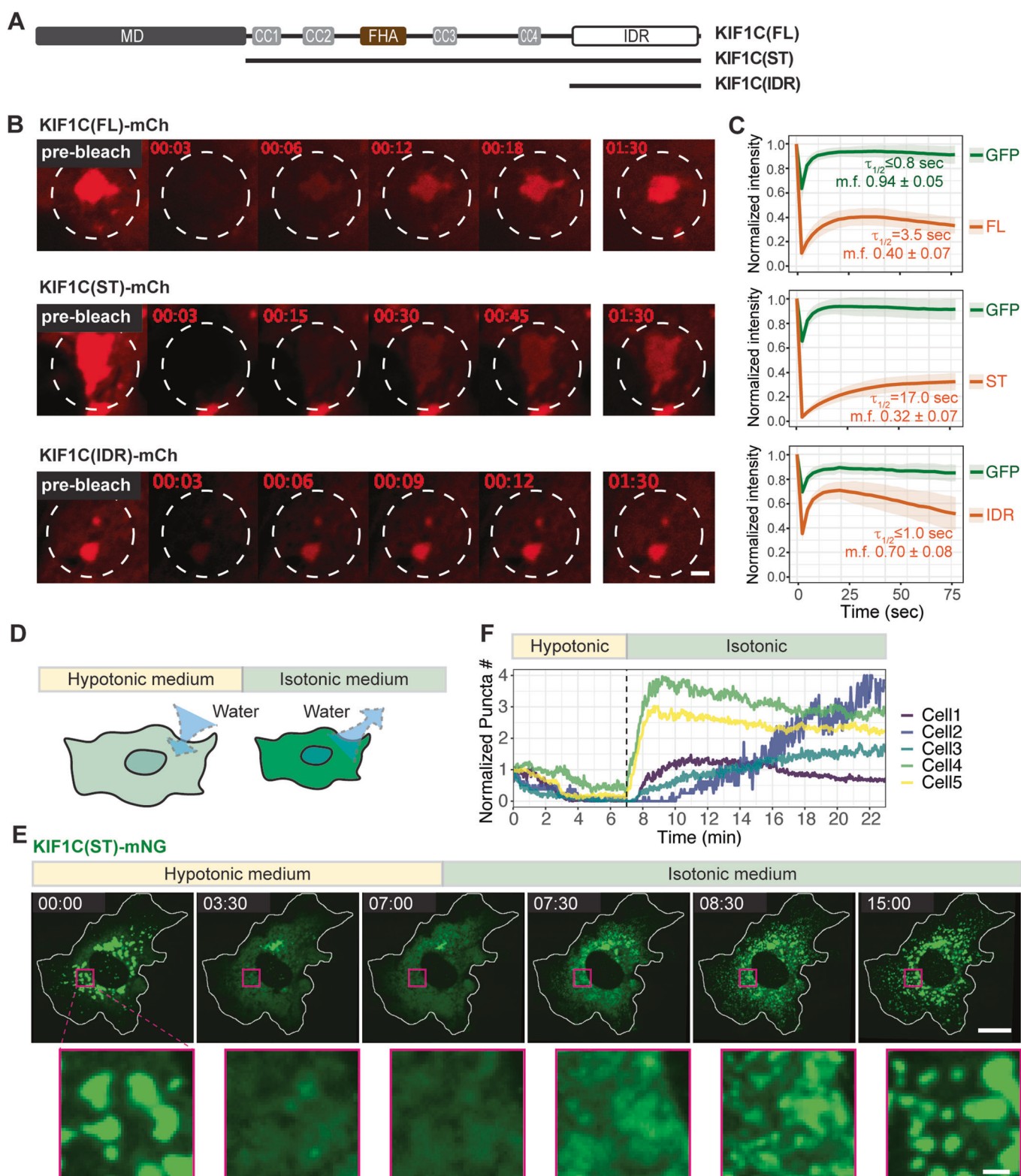

cells by transient transfection. GU-rich RNA oligos were incorporated into KIF1C(ST) and KIF1C(IDR) condensates in cells whereas polyA RNA oligos were not incorporated into condensates formed by either construct (Fig. 4B), suggesting that there is sequence selectivity for RNA incorporation into KIF1C

condensates. As a control, we carried out similar experiments with the related kinesin-3 KIF1Bα but found that KIF1Bα puncta did not incorporate either GU-rich or polyA RNAs (Appendix Fig. S6B).

Further evidence that KIF1C condensates are RNA granules comes from previously published proximity labeling experiments using BioID

**Figure 3. KIF1C puncta display properties of liquid condensates.**

(A) Schematic of constructs KIF1C(FL), KIF1C(ST), and KIF1C(IDR). (B) Representative images from live-cell imaging (see Movie EV2, 3, and 4) of mCherry-tagged KIF1C constructs (FL, ST, and IDR) expressed in COS-7 cells before and after photobleaching. White dashed lines indicate photobleached regions. Scale bar: 2 μm. Time label is [min:sec]. (C) Quantification of fluorescence intensity (solid line indicates mean, shaded area indicates SD) before and after photobleaching, normalized against the frame before photobleaching. For each cell, the fluorescence recovery of co-expressed EGFP was measured as a control. $N = 28$ cells for KIF1C(FL); $N = 20$ cells for KIF1C(ST); and $N = 9$ cells for KIF1C(IDR). The quantified recovery half-time ($\tau_{1/2}$) and mobile fraction (m.f.) are indicated in each plot. (D) Schematic of the cytoplasm dilution assay. (E) Representative images of mNG-tagged KIF1C(ST) localization before treatment, during hypotonic treatment, and upon return to isotonic media (see Movie EV5) in the cytoplasm dilution assay. Scale bar: 20 μm for whole cell views, 2 μm for magnified images. Time label is [min:sec]. (F) Quantification of change of puncta number over time in the cytoplasm dilution assay. x-axis: time, with a vertical dashed line indicating the time point of switching from hypotonic media to isotonic media. y-axis: puncta number normalized against the first frame of live-cell imaging. The example cell in (E) is Cell5 in the plot. Source data are available online for this figure.

fused to the KIF1C C-terminal tail domain (Kendrick et al, 2019). Examining their results, we found that nearly half of the proteins identified in the KIF1C-BioID interactome are RBPs, many of which are involved in mRNA processing (Appendix Fig. S6C). This suggests that KIF1C condensates likely enrich various RBPs, and may play a role in the regulation of mRNA processing during cell activities such as migration (Moissoglu et al, 2020; Theisen et al, 2012).

## KIF1C condensates enrich specific mRNAs in cell protrusions

To test whether the KIF1C IDR and LLPS are required for the localization of specific mRNAs, we generated KIF1C knockout (KO) cells using CRISPR-Cas9 gene editing of hTERT-hRPE1 cells whose genome is more stable than COS-7 cells. Three KO cell lines were selected and all three contained a frameshift mutation in amino acid 6, resulting in a premature stop codon and undetectable protein levels by western blotting (Appendix Fig. S7A,B). We confirmed that all three KIF1C KO clones display a defect in the distribution of *RAB13* mRNA to cell protrusions, as visualized by single-molecule fluorescence in situ hybridization (smFISH) (Fig. 5A,B), consistent with previous studies using RNAi to deplete KIF1C expression (Pichon et al, 2021). KIF1C KO cells also showed a defect in localization of *NET1* mRNA, another mRNA reported to be transported by KIF1C to cell protrusions, whereas the localization of *CAM1* mRNA, an mRNA not associated with KIF1C, was not affected (Appendix Fig. S7C,D).

We then expressed fluorescently-tagged KIF1C(FL) and KIF1C(ΔIDR) in KIF1C KO cells. KIF1C(FL)-mNG localized diffusely in cells and in puncta at the cell periphery (Fig. 5C), due to its ability to undergo microtubule-based motility and LLPS. The KIF1C(FL)-mNG condensates showed strong colocalization with *RAB13* mRNA (Fig. 5C,E) and were able to efficiently rescue *RAB13* mRNA localization to the cell periphery (Fig. 5F). In contrast, although KIF1C(ΔIDR)-mNG localized to the cell periphery due to its ability to undergo microtubule-based motility, it showed less colocalization with *RAB13* mRNA (Fig. 5D,E) and was less efficient at rescuing *RAB13* mRNA at the cell periphery (Fig. 5F). KIF1C(FL)-mNG was also able to selectively enrich *NET1* mRNA at cell protrusions whereas KIF1C(ΔIDR) was not (Appendix Fig. S7E,F). Together, these data indicate that KIF1C condensates can selectively incorporate RNAs and that KIF1C is involved in the targeting of these mRNAs to the cell periphery.

## A subregion of the KIF1C IDR is critical for enrichment of RAB13 mRNA in condensates

These results also indicate that the IDR is responsible for both LLPS and incorporation of select RNAs. To test whether KIF1C

LLPS and RNA binding are interdependent or two separate processes, we sought to distinguish sequences for LLPS and RNA binding within the IDR. We first hypothesized that the prion-like domain (PLD), a type of low-complexity region frequently found in RNA-binding proteins (RBPs) (Mészáros et al, 2018; Lancaster et al, 2014) would be involved in enrichment of *RAB13* mRNA into the condensates. However, deletion of the PLD [ST(ΔPLD), deletion of aa 1054-1080] or mutation of all glutamine and asparagine amino acids to alanines [ST(PLDmut)] (Fig. 6A,B) had no effect on condensate formation or the incorporation of *RAB13* mRNA (Fig. 6C,F).

We thus generated larger deletions across the IDR region (Fig. 6A,B). Deleting the first or last third of the IDR [constructs ST(ΔIDR1) or ST(ΔIDR3), respectively] had no effect on condensate formation or *RAB13* mRNA incorporation (Fig. 6D,F). In contrast, deleting the middle third of the IDR [construct ST(ΔIDR2)] altered the ability to undergo LLPS, with condensates forming in only ~30% of the transfected cells, and the condensates that did form appearing fewer in number and larger in size (Fig. 6D,F). Importantly, the ST(ΔIDR2) condensates that did form were unable to incorporate *RAB13* mRNA (Fig. 6D). This result suggests that RNA binding and LLPS of the KIF1C IDR are separate processes: the IDR2 region is responsible for RNA binding whereas multivalent interactions along the IDR are responsible for LLPS. Furthermore, condensates that fail to incorporate RNA are more liquid-like, as they readily fuse into larger puncta.

We further generated two smaller deletions within IDR2. Deleting the first half of IDR2 [ST(ΔIDR2a), Fig. 6A,B] resulted in the formation of fewer and larger condensates in the expressing cells and a failure to incorporate *RAB13* mRNA (Fig. 6E,F), similar to the full IDR2 deletion. In contrast, deleting the second half [ST(ΔIDR2b), Fig. 6A,B] did not affect condensate formation or mRNA incorporation (Fig. 6E,F). These results identify a segment of 48 amino acids (IDR2a, aa 938-985) to be the region responsible for RNA incorporation into KIF1C condensates.

## The KIF1C IDR is sufficient for driving LLPS in vitro

To confirm that the KIF1C IDR is sufficient for LLPS and RNA binding, we purified mNG-tagged KIF1C(CC4 + IDR) (aa 785–1103) and KIF1C(IDR) (aa 886–1103) proteins from *E. coli* (Fig. 7A; Appendix Fig. S8A). Both proteins were purified under high salt conditions (500 mM NaCl) and then underwent phase separation in vitro upon lowering the NaCl concentration to 100 mM (Fig. 7B). Both KIF1C proteins formed condensates at nM concentrations and in the absence of crowding agents whereas purified mNG did not form visible structures under the same conditions (Fig. 7B). Interestingly, the KIF1C(CC4 + IDR) droplets

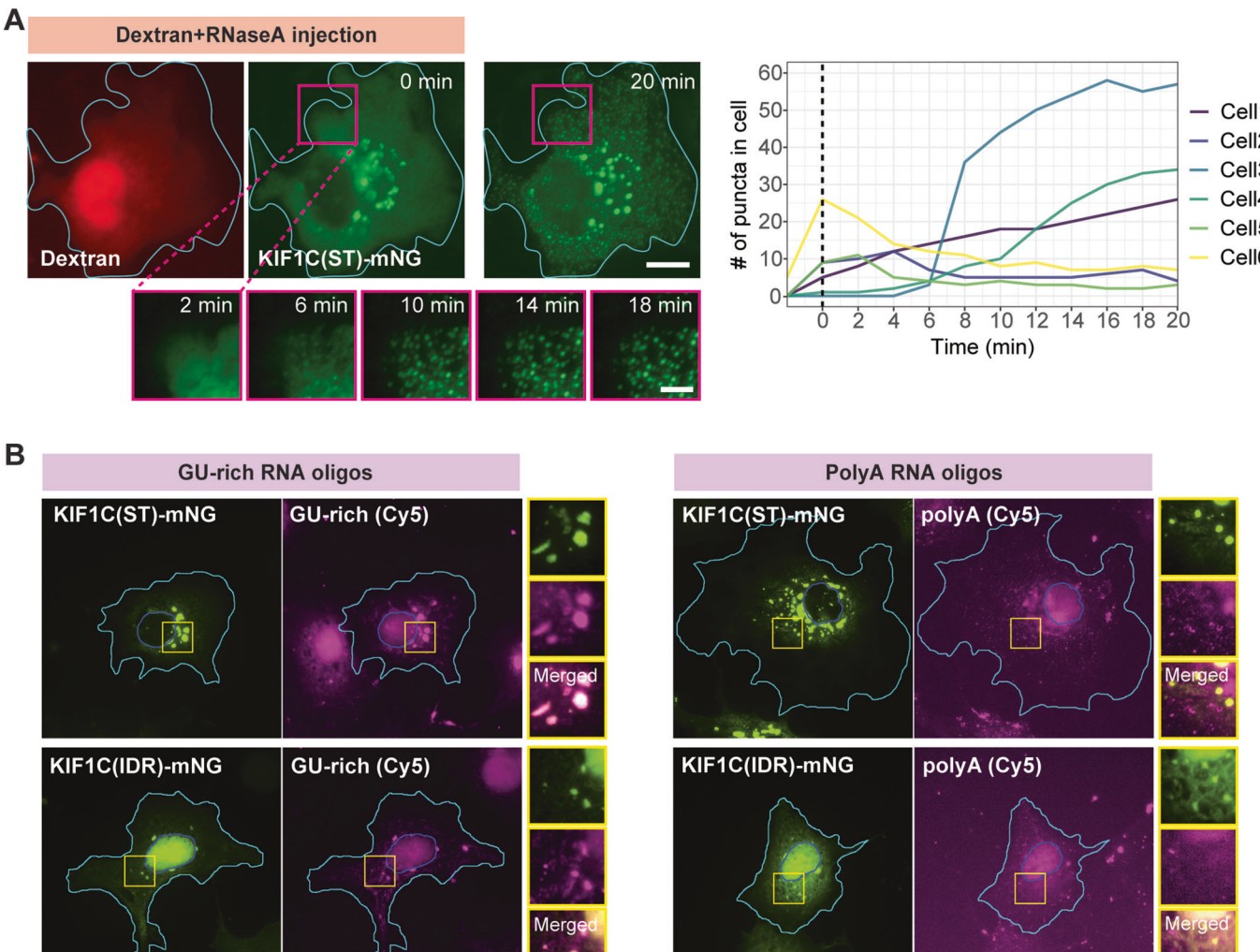

**Figure 4. KIF1C condensates display properties of RNA granules.**

(A) Representative images of a COS-7 cell expressing KIF1C(ST)-mNG before and after microinjection of RNaseA mixed with fluorescent dextran (see Movie EV7). Scale bar: 20 μm for whole cell views, 5 μm for magnified images. Cyan lines indicate cell boundaries. The plot on the right shows quantification of the number of puncta in a 25 μm × 25 μm area (magenta box) over time. y-axis, number of KIF1C(ST) puncta. x-axis, time after injection. The vertical dashed line represents the time point of microinjection. The example cell is Cell3 in the plot. (B) Fluorescently-labeled (Cy5) GU-rich or polyA RNA oligos were introduced into COS-7 cells expressing KIF1C(ST)-mNG (top row) or KIF1C(IDR)-mNG (bottom row). Representative images are shown. Cyan lines indicate cell boundaries. Blue lines indicate nuclear boundaries. Yellow boxes indicate the regions in the magnified images to the right. Scale bar: 20 μm for whole cell views, 5 μm for magnified images. Source data are available online for this figure.

were larger and more round in shape than the KIF1C(IDR) droplets (Appendix Fig. S8B), suggesting that inclusion of the structured CC4 region makes the droplets more liquid-like.

We determined the phase separation diagram for LLPS at different NaCl concentrations and found that at 100 mM NaCl buffer, which has an ionic strength close to the physiological range, the critical concentrations for LLPS were ~62.5 nM for KIF1C(CC4 + IDR) and ~20 nM for KIF1C(IDR) (Fig. 7C). Interestingly, these values are close to the estimated concentration of endogenous KIF1C [25.5 ± 7.6 nM, (mean ± SD), Appendix Fig. S8C], indicating that local concentration of endogenous KIF1C molecules may be sufficient to push LLPS over the critical concentration. In addition, the KIF1C(CC4 + IDR) condensates were observed to undergo typical fusion behavior (Fig. 7D; Movie

EV9). Overall, these in vitro results demonstrate that the KIF1C IDR is sufficient to drive LLPS and that KIF1C LLPS occurs at concentrations close to endogenous levels.

To confirm that the KIF1C IDR is sufficient for RNA binding, various concentrations of fluorescently-labeled RNA oligos were mixed with 100 nM KIF1C(CC4 + IDR)-mNG in 100 mM NaCl buffer. At high concentrations (μM) of either GU-rich or polyA RNAs, the formation of KIF1C(CC4 + IDR)-mNG condensates was inhibited (Fig. 7E). This is consistent with the RNaseA injection experiments suggesting that LLPS can be suppressed by high concentrations of RNAs (Fig. 4A). At medium concentrations (120–600 nM), both RNA oligos became enriched in the CC4 + IDR condensates, whereas at lower concentrations (15–60 nM), the GU-rich oligos were selectively enriched in the condensates (Fig. 7E,F), suggesting sequence

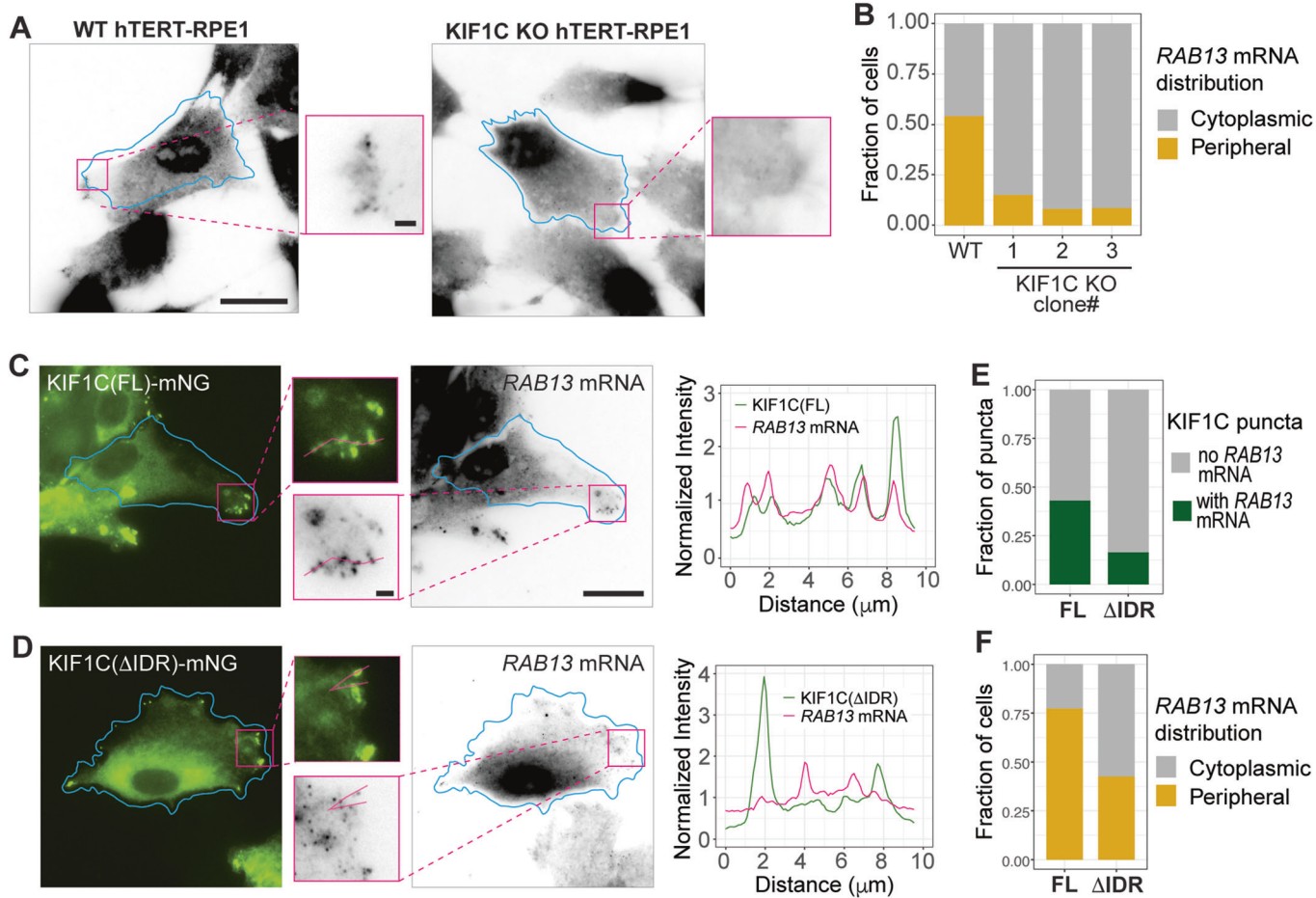

**Figure 5. KIF1C condensates enrich endogenous *RAB13* mRNA at cell protrusions.**

(A) Representative smFISH images showing the distribution of endogenous *RAB13* mRNA in WT (left) and KIF1C KO (right) hTERT-RPE1 cells. Scale bar: 20 μm for whole cell views, 2 μm for magnified images. Cyan lines indicate cell boundaries. (B) Quantification of the fraction of cells displaying cytoplasmic or peripheral distribution of *RAB13* mRNA in WT cells and three KIF1C KO clones. $N = 72$ cells for WT, $N = 66$ cells for KO clone #1, $N = 73$ cells for KO clone #2, and $N = 69$ cells for KO clone #3. y-axis: fraction of cells showing peripheral or cytoplasmic distribution of *RAB13* mRNA. (C–E) Rescue of *RAB13* mRNA enrichment and localization in KIF1C KO clone #1 cells. (C, D) Representative images of expressed KIF1C(FL)-mNG (C) or KIF1C(ΔIDR)-mNG (D) and smFISH for endogenous *RAB13* mRNA. Cyan lines indicate cell boundaries, magenta boxes indicate the regions of the magnified imaged in the middle. Magenta lines in the magnified images indicate line scan for intensity profiles. Scale bar: 20 μm for whole cell views, 2 μm for magnified images. (Right) Fluorescence intensity profiles of KIF1C protein (green lines) and *RAB13* mRNA (magenta lines). x-axis, distance along the scanning line. y-axis, intensity values normalized to the average intensity along the entire line. (E) Quantification of the fraction of cells with peripheral or cytoplasmic distribution of *RAB13* mRNA in cells expressing KIF1C(FL)-mNG ($N = 84$ cells) or KIF1C(ΔIDR)-mNG ($N = 68$ cells). (F) Quantification of the fraction of KIF1C puncta with or without *RAB13* mRNA enrichment for KIF1C(FL)-mNG ($N = 64$ cells) and KIF1C(ΔIDR)-mNG ($N = 63$ cells). Source data are available online for this figure.

selectivity. These results are consistent with the selective incorporation of GU-rich oligos into KIF1C(ST) condensates in cells (Fig. 4B).

## Endogenous KIF1C forms condensates at the cell periphery

Overexpression studies are useful for probing the mechanism of KIF1C condensate formation and for understanding condensate properties, however, we wanted to verify that KIF1C forms condensates under physiological conditions. We stained WT and KIF1C KO cells with an antibody raised against an epitope in the tail domain (aa 850–950) of KIF1C (Fig. 8A). Although the antibody shows a high non-specific background signal throughout the cytoplasm and nucleoplasm of both WT and KO cells (Fig. 8B), small puncta could be observed only at the periphery of WT cells (Fig. 8B,C). To further verify the specificity of the antibody, we used the KIF1C KO cells to generate stable cell

lines expressing low levels of KIF1C(FL)-mNG (Movie EV10) or KIF1C(ΔIDR)-mNG. The antibody only stains small KIF1C puncta at the periphery of cells expressing KIF1C(FL)-mNG (Fig. 8D). The puncta of endogenous KIF1C protein in WT cells incorporate endogenous *RAB13* mRNA (Fig. 8E,F), indicating that the puncta are RNA granules.

# Discussion

Our findings demonstrate that the kinesin-3 motor protein KIF1C undergoes LLPS to form biomolecular condensates in the cytoplasm of mammalian cells. The IDR in the KIF1C tail domain is necessary for the formation of dynamic, liquid-like puncta in cells and sufficient for LLPS in vitro. KIF1C's LLPS behavior enables it to function in RNA

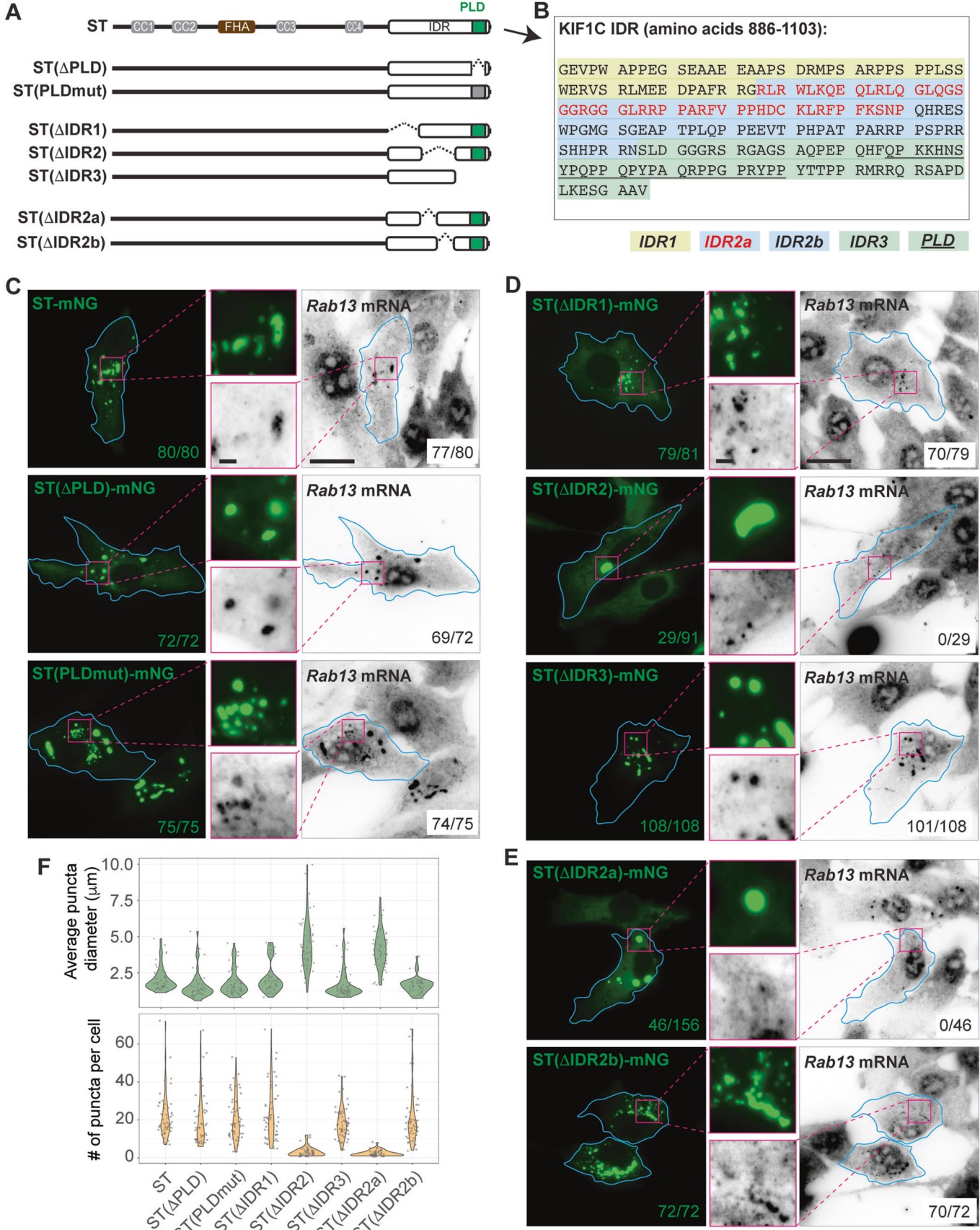

◄ **Figure 6. KIF1C IDR subregion required for *RAB13* mRNA localization.**

(A) Schematic of constructs KIF1C(ST), ST(ΔPLD), ST(PLDmut), ST(ΔIDR1), ST(ΔIDR2), ST(ΔIDR3), ST(ΔIDR2a), and ST(ΔIDR2b). The PLDmut construct has mutations N1059A, Q1063A, Q1066A, and Q1071A. (B) The amino acid sequence of KIF1C IDR, with subregions annotated. (C–E) Localization of *RAB13* mRNA in KIF1C KO clone #1 cells expressing (C) deletion or mutation of the PLD, (D) deletion of subregions of the IDR, and (E) deletion of subregions within IDR2. Cyan lines indicate cell boundaries; magenta boxes indicate the regions in the magnified images in the middle. Scale bar: 20 μm for whole cell views, 2 μm for magnified images. Numbers in green text: number of cells with puncta; numbers in black text: number of cells with *RAB13* mRNA enrichment in the KIF1C condensates. (F) Quantification of puncta diameter (top) and puncta number (bottom) for each construct. Each dot represents one cell. For ST: 21.8 ± 12.0 puncta per cell (mean ± SD), 2.06 ± 0.83 μm diameter (mean ± SD), $N = 49$ cells; For ST(ΔPLD): 20.1 ± 13.7 puncta per cell, 1.62 ± 0.94 μm diameter, $N = 47$ cells; For ST(PLDmut): 20.8 ± 11.1 puncta per cell, 1.83 ± 0.89 μm diameter, $N = 56$ cells; For ST(ΔIDR1), 23.1 ± 14.3 puncta per cell, 2.13 ± 1.04 μm diameter, $N = 53$ cells; For ST(ΔIDR2), is 3.1 ± 2.4 puncta per cell, 4.44 ± 1.74 μm diameter, $N = 51$ cells; For ST(ΔIDR3), 18.4 ± 8.1 puncta per cell, 1.71 ± 0.98 μm diameter, $N = 62$ cells; For ST(ΔIDR2a), 2.4 ± 1.5 puncta per cell, 4.15 ± 1.28 μm diameter, $N = 51$ cells; For ST(ΔIDR2b), 20.9 ± 14.1 puncta per cell, 1.63 ± 0.55 μm diameter, $N = 53$ cells. Source data are available online for this figure.

localization via selective recruitment of mRNA molecules into the KIF1C condensates and their localization at cell protrusions.

Given previous work on KIF1C's roles in mRNA transport and cell migration (Moissoglu et al, 2019, 2020; Pichon et al, 2021), we propose that KIF1C drives microtubule-based transport of specific mRNAs to the cell periphery and that its accumulation at microtubule plus ends initiates the formation of biomolecular condensates that accumulate mRNAs and regulate the concentration and/or activity of the encoded proteins at cell protrusions (Fig. 9). Thus, the KIF1C condensates may represent a new type of RNA granule. Future investigations will help us understand how KIF1C LLPS regulates the binding of specific mRNAs and their processing, translation, and stability in cells, and how those protrusion-localized condensates affect cellular activities.

## LLPS of kinesins and other microtubule-associated proteins

We demonstrate that the IDR in the C-terminal tail domain of the kinesin-3 KIF1C is necessary and sufficient for LLPS and formation of a biomolecular condensate with liquid-like properties. KIF1C is the first kinesin, to our knowledge, demonstrated to undergo LLPS despite the fact that many of the 45 kinesin genes in the human genome contain significant regions of intrinsically disordered residues (Seeger and Rice, 2013). Recent work by Maan et al, shows that the *S. pombe* kinesin molecule Tea2 can enter LLPS droplets in cells and in vitro under crowding conditions (Maan et al, 2023), but whether Tea2 drives LLPS on its own in an IDR-dependent manner is not known. Interestingly, both KIF1C and Tea2 can drive motility of condensates along microtubules.

Within the kinesin-3 family, KIF14 contains a ~350 aa segment N-terminal to its motor domain which is predicted to be intrinsically disordered. Although phase separation of the purified KIF14 IDR has not been tested, the KIF14 IDR does not form puncta when expressed in mammalian cells (Gruneberg et al, 2006), suggesting that it does not undergo LLPS. Rather, KIF14's IDR has been shown to bind partner proteins such as PRC1 and actin and to regulate KIF14 motility (Samwer et al, 2013; Zhernov et al, 2020). Of the other kinesin-3 family members, only the KIF1Bα tail domain shows a high score in IDR and PLD predictions (Appendix Fig. S2C). KIF1Bα is closely related to KIF1C as they share ~62% amino acid sequence identity overall, and ~33% identity within their IDR domains (KIF1Bα aa 870–1153, KIF1C aa 886–1103). A KIF1Bα stalk+tail construct, KIF1Bα(ST), forms small puncta when expressed in hTERT-RPE cells but forms a network-like structure when expressed in COS-7 cells (Appendix Fig. S5B), suggesting that KIF1Bα may also be capable of LLPS. However, KIF1Bα puncta do not associate with RNA-containing P-bodies and stress granules (Appendix Fig. S5C) or recruit exogenous RNA oligos (Appendix Fig. S6B).

Microtubules have emerged as a platform for LLPS (Volkov and Akhmanova, 2023) as several non-motor microtubule-associated proteins (MAPs) can undergo LLPS, including tau, EB1, CLIP-170, BuGZ, TPX2, and CAMSAP2, and thereby regulate microtubule nucleation, branching, bundling, and/or mitotic spindle formation (Tan et al, 2019; Siahaan et al, 2019; Wu et al, 2021; Miesch et al, 2023; Jiang et al, 2015; Imasaki et al, 2022; Ambadipudi et al, 2017; King and Petry, 2020; Maan et al, 2023; Hernández-Vega et al, 2017; Setru et al, 2021; Song et al, 2023). However, KIF1C condensates appear to be distinct from these other microtubule-associated biomolecular condensates. For example, although both KIF1C and CLIP-170 undergo LLPS at microtubule plus ends, KIF1C condensates do not recruit free tubulin (Appendix Fig. S4), while RNA molecules are excluded from CLIP-170 condensates (Wu et al, 2021).

## The KIF1C IDR and coiled-coil domains influence the rheological properties of the condensate

We demonstrate that the C-terminal KIF1C IDR is essential for LLPS as constructs that lack the IDR do not form droplets in mammalian cells whereas constructs that contain only the IDR form droplets in cells and in vitro. We find that the purified IDR protein readily undergoes LLPS at physiological salt conditions, in the absence of crowding agents, and near the estimated KIF1C endogenous concentration. This is in contrast to other MAPs that accumulate at microtubule plus ends, such as EB1 and CLIP-170, which undergo LLPS at μM concentrations or in the presence of crowding agents (Miesch et al, 2023; Maan et al, 2023; Song et al, 2023).

Several aspects of KIF1C's IDR and resulting LLPS are reminiscent of findings on prion-like RNA-binding proteins (RBPs), such as FUS, TDP-43, and TAF15. First, KIF1C and the prion-like RBPs show robust IDR-dependent phase separation in the absence of crowding agents at low protein concentrations and physiologically-relevant salt concentrations (Lin et al, 2016; Pak et al, 2016; Wang et al, 2018). Second, RNA can regulate phase separation of KIF1C and the prion-like RBPs (Schwartz et al, 2013) as the LLPS behavior of KIF1C and prion-like RBPs is buffered by the non-selective RNA pool (Maharana et al, 2018; Duan et al, 2022). RNA can also change the rheological properties of KIF1C condensates, as failure to incorporate selective RNAs makes the condensates more liquid-like, similar to other RNA granules (Lin et al, 2015; Roden and Gladfelter, 2021). Third, KIF1C and the prion-like RBPs share sequence similarity in their IDRs. For the prion-like RBPs, phase separation derives from collective interactions between R residues in the RBD and Y residues in the PLD (Wang et al, 2018). For KIF1C, the RBD (IDR2a) is enriched in R residues and modulates phase separation whereas the PLD (in IDR3) is enriched in Y residues but is dispensable for phase separation. Future studies are needed to understand the interplay between RNA, the RBD, and the PLD in regulating the phase behavior of KIF1C.

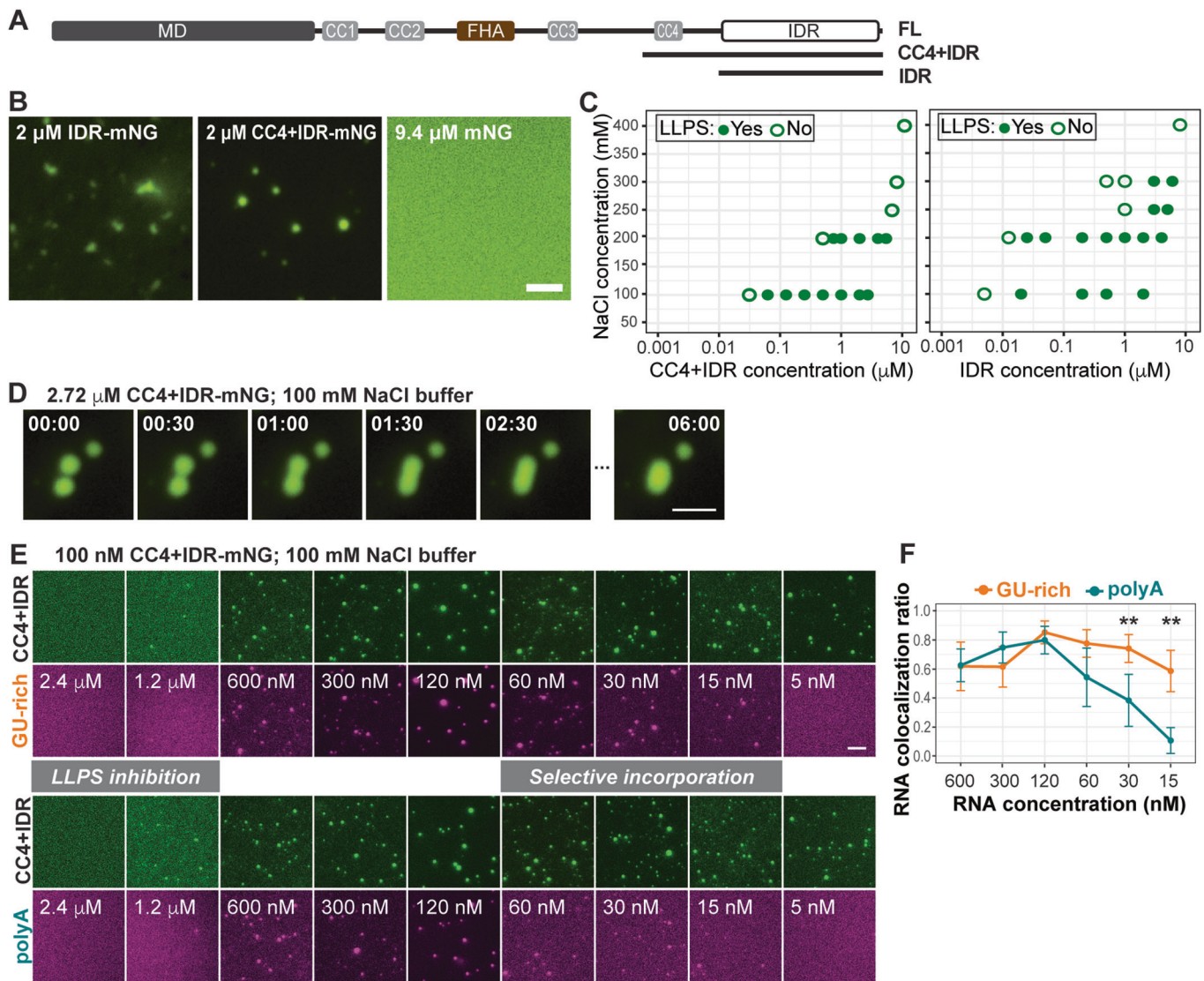

**Figure 7. The IDR is sufficient for LLPS and RNA incorporation in vitro.**

(A) Schematic of the mNG-tagged KIF1C truncations CC4 + IDR and IDR. (B) Purified IDR-mNG and CC4 + IDR-mNG proteins undergo LLPS whereas purified mNG does not (buffer: 100 mM NaCl). Scale bar: 2 μm. (C) Phase diagram of condensate formation for CC4 + IDR-mNG (left) and IDR-mNG (right) at different salt concentrations. x-axis: protein concentration plotted on log scale; y-axis: NaCl concentrations. Solid green dots indicate that LLPS was observed under given conditions, open green circles indicate that LLPS was not observed under given conditions. (D) Representative images of a fusion event observed for CC4 + IDR condensates (see Movie EV9). Scale bar: 2 μm. Time label is [min:sec]. (E, F) Incorporation of fluorescently-labeled (Cy5) GU-rich or polyA RNA oligos into CC4 + IDR-mNG condensates (buffer: 100 mM NaCl). (E) Representative images for one experiment of 4 independent experiments. Scale bar: 5 μm. (F) Quantification of RNA colocalization with CC4 + IDR-mNG condensates. y-axis: RNA and condensate colocalization ratio (mean ± SD) calculated across ~60 field of views in 4 independent experiments. **$p < 0.02$ (t-test). Source data are available online for this figure.

We also find that although the C-terminal KIF1C IDR contains the requisite sequence features to drive phase separation, inclusion of the adjacent structured CC4 segment can tune the rheological properties of the resulting condensates. In vitro, droplets formed by purified KIF1C IDR protein are irregular in shape and do not undergo fusion whereas droplets formed by purified KIF1C(CC4 + IDR) protein are round in shape, undergo fission and fusion, and require a higher concentration for phase separation. These results suggest that KIF1C(CC4 + IDR) condensates are more liquid-like, indicating that inclusion of the structured domain makes the condensate more fluid. These findings indicate that LLPS of KIF1C fits within the theory of associative polymers, or the sticker-and-

spacer theory, in which the IDR is mostly composed of the associative motifs, or stickers, that provide the multivalency for molecular interactions and is the main component determining the phase separation behavior whereas structured domains like CC4 act as spacers, which impact the flexibility of molecules and tune the rheological properties of condensates (Rubinstein and Semenov, 1998; Wang et al, 2018; Choi et al, 2020).

## Mechanisms of mRNA binding by kinesins

RNA transport requires microtubule-dependent motor proteins and several kinesin family members in addition to KIF1C have been

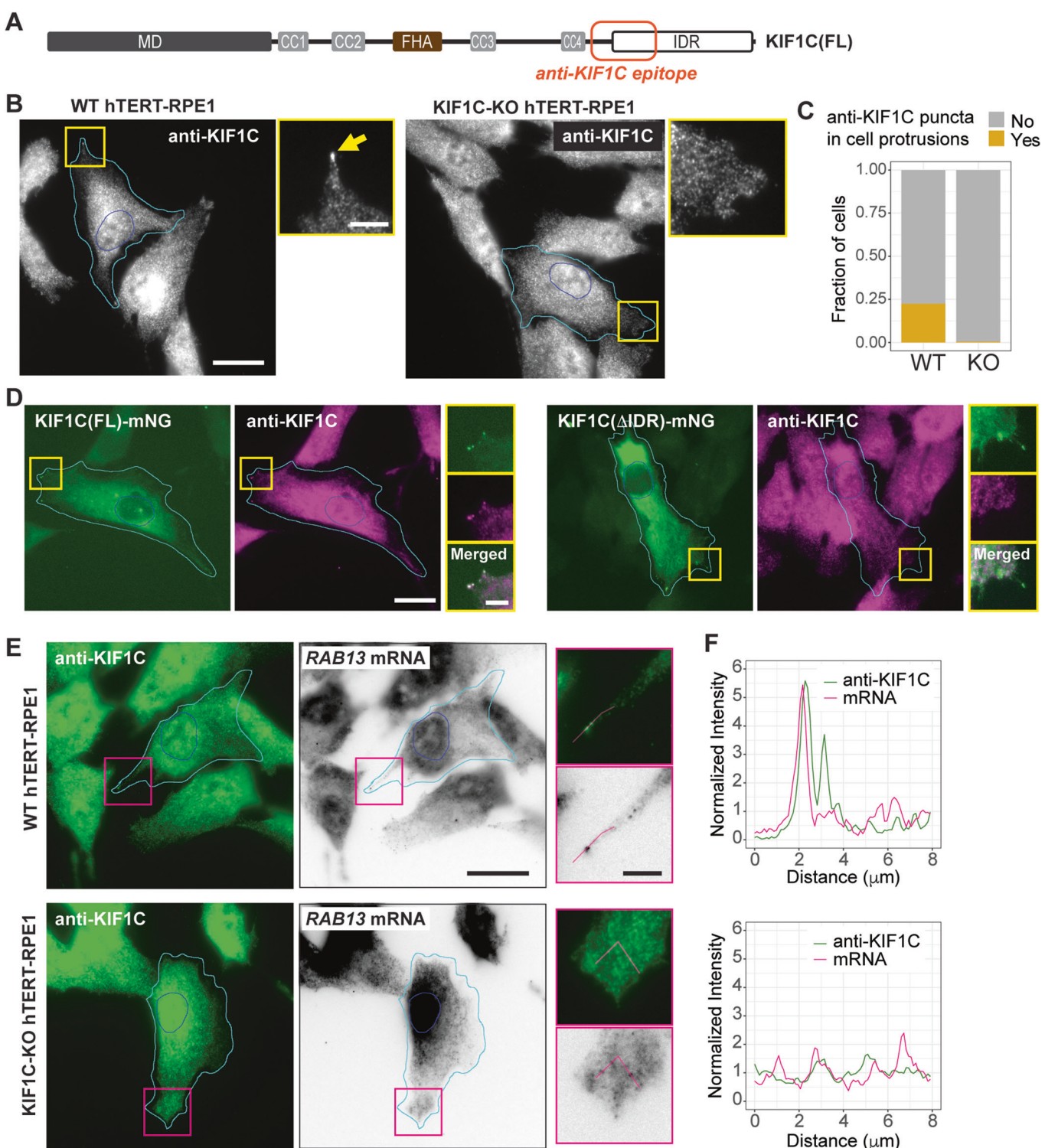

implicated (Fernandopulle et al, 2021; Dalla Costa et al, 2021; Das et al, 2021). An outstanding question in the field is the molecular mechanisms by which motors associate with mRNA and other RNAs. One possibility is that kinesins can bind directly to RNAs. For KIF1C, this possibility is supported by recent work demonstrating that KIF1C can undergo direct binding to RNAs in the EJC (Nagel et al, 2022). Furthermore, KIF1C was the only kinesin

identified in two studies that used UV cross-linking methods to capture RNA-binding proteins in an unbiased manner (Baltz et al, 2012; Castello et al, 2012). Our work extends these findings to show that the purified KIF1C(CC4 + IDR) can bind to RNA oligos in vitro in a sequence-selective manner.

An alternative possibility is that RBPs serve as adapter proteins to link kinesins to specific mRNA cargoes in cells. This possibility fits with the

**Figure 8. KIF1C forms biomolecular condensates at endogenous expression level.**

(A) Schematic showing the location of anti-KIF1C epitope. (B) Representative images of WT and KIF1C KO hTERT-RPE1 cells stained with an antibody against KIF1C. Cyan lines indicate cell boundaries. Blue lines indicate nuclear boundaries. The yellow arrow points to a cell protrusion with a KIF1C punctum. Yellow boxes indicate the regions magnified on the right. Scale bar: 20 μm for whole cell views, 5 μm for magnified images. (C) Quantification of fraction of WT and KIF1C-KO hTERT-RPE1 cells containing KIF1C puncta in cell protrusions. $N = 439$ WT cells and 361 KIF1C-KO cells. (D) Immunostaining of KIF1C in KO cells stably expressing KIF1C(FL)-mNG or KIF1C(ΔIDR)-mNG. Cyan lines indicate cell boundaries. Blue lines indicate nuclear boundaries. Yellow boxes indicate the regions magnified on the right. Scale bar: 20 μm for whole cell views, 5 μm for magnified images. (E, F) Colocalization of *RAB13* mRNA with endogenous KIF1C in cell protrusions. (E) Representative images of WT and KO cells stained with anti-KIF1C antibody and *RAB13* mRNA smFISH probes. Cyan lines indicate cell boundaries. Blue lines indicate nuclear boundaries. Magenta boxes indicate the regions magnified on the right. The magnified images show the location of the line scans for intensity profiles shown in (F). Scale bar: 20 μm for whole cell views, 5 μm for magnified images. (F) Fluorescence intensity profiles of KIF1C immunofluorescence (green lines) and *RAB13* mRNA (magenta lines). x-axis, distance along the scanning line. y-axis, intensity values normalized to the average intensity along the entire line. Source data are available online for this figure.

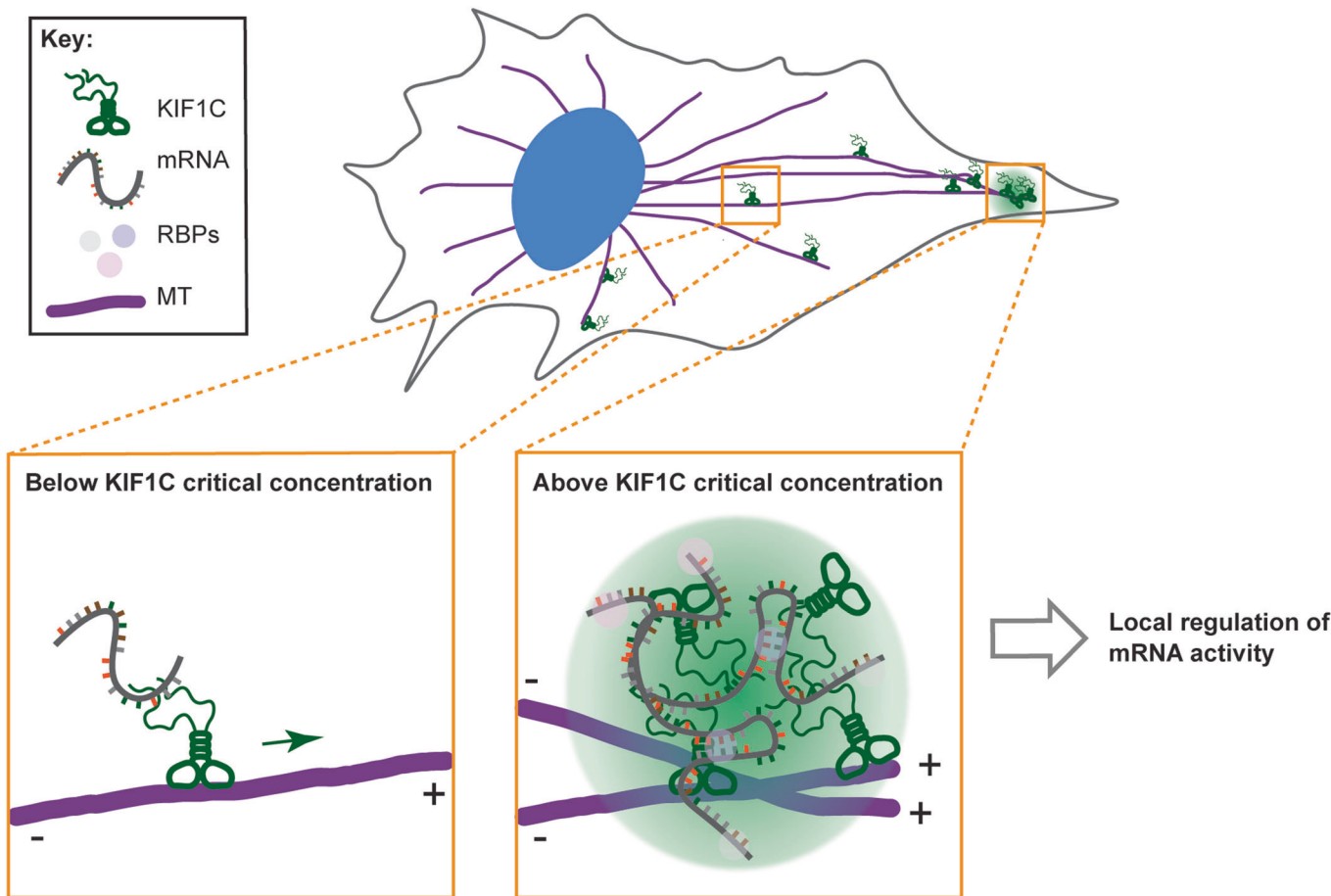

**Figure 9. Working model of KIF1C condensate formation during mRNA transport.**

KIF1C drives microtubule-based transport of specific mRNAs to the cell periphery, mostly as single motors. Upon reaching the microtubule plus ends in the cell periphery, KIF1C accumulates above the critical concentration of LLPS, driving the formation of biomolecular condensates that regulate mRNA activity and function during cellular processes such as migration.

classical kinesin-adapter-cargo model (Rodrigues et al, 2021) and has been shown for kinesin-1 and kinesin-2 which utilize adapter proteins to link to mRNA cargoes (Wu et al, 2020; Baumann et al, 2020). For KIF1C, this possibility is supported by recent work demonstrating that muscle blind-like (MBNL) serves as an adapter protein bridging KIF1C and mRNAs (Hildebrandt et al, 2023). It is important to note that these models of direct RNA binding and adapter-mediated RNA binding are not mutually exclusive and a hybrid model is also possible in which RBPs stabilize the direct interaction of kinesins with specific mRNAs, as has

been proposed for kinesin-1 and the adapter protein atypical tropomyosin 1 (Dimitrova-Paternoga et al, 2021; Vaishali et al, 2021).

## Relationship between KIF1C LLPS and mRNA transport

We show that the KIF1C IDR is required for both LLPS and delivery of *RAB13* and *NET1* mRNAs to the cell periphery. An open question is at what point during the mRNA transport cycle KIF1C undergoes LLPS. Given that the purified IDR undergoes LLPS near

the estimated KIF1C endogenous concentration, we propose that KIF1C is on the edge of LLPS in cells and its formation of RNA-containing condensates is spatially regulated. That is, we hypothesize that individual KIF1C proteins walk to the plus ends of microtubules in the cell periphery where their accumulation results in a high local concentration that allows LLPS on the microtubule through an IDR-based mechanism (Fig. 9). At concentrations above endogenous levels, whether by overexpression or concentration of the cytoplasm, KIF1C readily undergoes phase separation, and the resulting condensates display several biophysical characteristics consistent with a liquid-like state including round morphology, fusion and fission, and dynamic exchange of internal components with the surrounding solution.

Several lines of evidence support the proposal that KIF1C LLPS occurs at the end of the transport journey when the motor-cargo complex reaches the plus ends of microtubules (Fig. 9). First, we show that endogenous KIF1C protein forms only a few small droplets in the cell periphery, consistent with previous work (Norris and Mendell, 2023; Theisen et al, 2012). Second, mRNAs whose localization at the cell periphery is dependent on KIF1C can be transported in the form of single copies and then coalesce into larger clusters once they reach their destination in cell protrusions (Pichon et al, 2021; Moissoglu et al, 2019). Finally, individual KIF1C proteins are superprocessive and readily reach and accumulate at the plus ends of microtubules (Siddiqui et al, 2019; Kendrick et al, 2019).

An outstanding question is what effect KIF1C's LLPS and transport have on the processing, translation, and/or stability of the mRNAs it delivers to the cell periphery. Previous work demonstrated that mRNA in clusters at the tips of retracting cell protrusions are translationally silent (Moissoglu et al, 2019), suggesting that KIF1C condensates provide a platform for regulating RNA activity. Support for this model comes from proximity labeling experiments using BioID fused to the KIF1C C-terminal tail domain (Kendrick et al, 2019). Indeed, nearly half of the proteins identified in the KIF1C-BioID interactome are RBPs, many of which are involved in mRNA decay (Appendix Fig. S6C), suggesting an inhibitory role of KIF1C condensates in translational regulation during cell activities such as migration (Moissoglu et al, 2020; Theisen et al, 2012).

# Methods

### Reagents and Tools Table

| Reagent or Resource | Reference or Source | Identifier |
|---|---|---|
| **Antibodies** | | |
| anti-KIF1C (1: 1000 for WB; 1: 200 for IF) | Abcam | Cat# ab72238 |
| anti-GAPDH (1: 50,000 for WB) | Abcam | Cat# ab181602 |
| anti-G3BP (1: 500 for IF) | Abcam | Cat# ab56574 |
| anti-DDX6 (1: 500 for IF) | Bethyl Laboratories | Cat# A300-461A-T |
| anti-β-tubulin (1: 2000 for IF) | Developmental Studies Hybridoma Bank | Cat# E7; RRID: AB_528499 |
| anti-αTubulin (clone 12G10, 1: 1000 for WB) | Developmental Studies Hybridoma Bank | Cat# 12G10; RRID: AB_1157911 |

| Reagent or Resource | Reference or Source | Identifier |
|---|---|---|
| **Cell lines** | | |
| COS-7 cells | ATCC | RRID: CVCL_0224 |
| hTERT-RPE1 cells | ATCC | RRID: CVCL_4388 |
| KIF1C-KO hTERT-RPE1 cells | This study | N/A |
| Rosetta2 competent cells | Millipore Sigma | Cat# 71397 |
| **Recombinant DNA** | | |
| pN1-HsKIF1C-GFP | AddGene | Cat# 130977 |
| eSpCas9(1.1) | AddGene | Cat# 71814 |
| EGFP-RatCLIP170 | Gift from Ryoma Ohi | N/A |
| pN1-HsKIF1Bβ-mNG | This study | N/A |
| pN1-KIF16B-mCit | (Soppina et al, 2014) | N/A |
| pN1-KIF13B-3xmCit | (Soppina et al, 2014) | N/A |
| pC1-mCherry-Rab6 | Gift from Anna Akhmanova (Matanis et al, 2002) | N/A |
| pN1-HsKIF1C-mCherry | This study | N/A |
| pN1-HsKIF1C-Halo-FLAG | This study | N/A |
| pN1-HsKIF1C(1-376)-1Anc-GCN4-mCherry | This study | N/A |
| pN1-HsKIF1C(ΔIDR)-mCherry | This study | N/A |
| pN1-HsKIF1C(ST)-mCherry | This study | N/A |
| pN1-HsKIF1C(IDR)-mCherry | This study | N/A |
| pN1-HsKIF1C(ST)-Halo-FLAG | This study | N/A |
| pN1-HsKIF1C(IDR)-Halo-FLAG | This study | N/A |
| pN1-HsKIF1C(FL)-mNG | This study | N/A |
| pN1-HsKIF1C(ΔIDR)-mNG | This study | N/A |
| pN1-HsKIF1C(ST)-mNG | This study | N/A |
| pN1-HsKIF1C(IDR)-mNG | This study | N/A |
| pN1-HsKIF1C(1-376)-1Anc-GCN4-mNG | This study | N/A |
| pN1-HsKIF1C(785-1103)-mNG | This study | N/A |
| pN1-HsKIF1C(691-1103)-mNG | This study | N/A |
| pN1-HsKIF1C(595-1103)-mNG | This study | N/A |
| pN1-HsKIF1C(500-1103)-mNG | This study | N/A |
| pN1-HsKIF1Bα(355-1153)-mNG | This study | N/A |
| pMW96 | Addgene | Cat# 178066 |
| pMW96-His-SUMO-HsKIF1C(IDR)-mNG | This study | N/A |
| pMW96-His-SUMO-HsKIF1C(CC4 + IDR)-mNG | This study | N/A |

| Reagent or Resource | Reference or Source | Identifier |
|---|---|---|
| pMW96-His-SUMO-mNG | This study | N/A |
| pN1-HsKIF1C(1-784)-mNG | This study | N/A |
| pN1-HsKIF1C(1-690)-mNG | This study | N/A |
| pN1-HsKIF1C(1-594)-mNG | This study | N/A |
| pN1-HsKIF1C(1-499)-mNG | This study | N/A |
| **Software** | | |
| Fiji | (Schneider et al, 2012) | https://fiji.sc/ |
| R Statistical Software (v4.2.0) | | https://www.R-project.org/ |
| RStudio | RStudio Team (2023) | http://www.rstudio.com/ |
| **Other Reagents** | | |
| Synthesized GU-rich RNA oligos (Cy5-(GU)6) | MilliporeSigma | N/A |
| Synthesized polyA RNA oligos (Cy5-A13) | MilliporeSigma | N/A |
| TEV enzymes | Ryoma Ohi lab | N/A |
| HisPur Ni-NTA Spin Columns, 0.2 mL | Thermo Scientific | Cat# 88224 |
| Vivaspin 500, 10 kDa MWCO | Millipore Sigma | Cat# GE28-9322-25 |
| Primary smFISH probes | Custom order from Integrated DNA Technologies (IDT) | N/A |
| Secondary smFISH probes | Custom order from Integrated DNA Technologies (IDT) | N/A |

## Molecular cloning

A plasmid for expression of full-length human KIF1C (pKIF1C-GFP) was a gift from Anne Straube [Addgene plasmid #130977 (Efimova et al, 2014)]. All truncated versions of KIF1C (see Resource Table), unless otherwise stated, were constructed by PCR (Q5 High-Fidelity DNA Polymerase, NEB cat#M0491S) and ligation or Gibson isothermal assembly (NEBuilder HiFi DNA Assembly Master Mix, cat# E2621S). To obtain a constitutively-active, dimeric KIF1C motor domain [KIF1C(MD)-LZ-mNG], a leucine zipper (LZ) motif was included to ensure dimerization as truncated kinesin-3 motors can form weak dimers, as shown in previous studies (Soppina et al, 2014). Plasmids for expression of fusion proteins in mammalian cells utilize the cytomegalovirus (CMV) promoter. Fragments for bacterial expression were cloned into pMW96 plasmid (gift of Tarun Kapoor, Addgene plasmid #178066) which uses the T7 promoter to drive protein expression in Rosetta2 *E. coli* cells. All plasmids were verified by DNA sequencing.

## Protein structure prediction

The intrinsically disordered region (IDR) was predicted using IUPred2 (https://iupred2a.elte.hu). The prion-like domain (PLD) was predicted using PLAAC (http://plaac.wi.mit.edu/) with parameters set to core length = 60, α = 100. Results from IUPred2 (ANCHOR score) and PLAAC (HMM.PrD-like score) were visualized in the same plots by custom scripts in R (https://cran.r-project.org).

## Cell culture and transient transfection

COS-7 [male *Ceropithecus aethiops* (African green monkey) kidney fibroblast, RRID: CVCL_0224] cells were grown in Dulbecco's modified Eagle medium (Gibco) supplemented with 10% (vol/vol) Fetal Clone III (HyClone) and 2 mM GlutaMAX (L-alanyl-L-glutamine dipeptide in 0.85% NaCl, Gibco). hTERT-RPE1 cells (female *Homo sapiens* retinal pigment epithelium, RRID: CVCL_4388) were grown in DMEM/F12 with 10% (vol/vol) FBS (HyClone), 0.01 mg/ml hygromycin B, and 2 mM GlutaMAX (Gibco). All cell lines were purchased from American Type Culture Collection and grown at 37 °C with 5% (vol/vol) $CO_2$. All cell lines are checked annually for mycoplasma contamination and COS-7 cells were authenticated through mass spectrometry (the protein sequences exactly match those in the *Ceropithecus aethiops* genome).

For transfection of plasmid DNA in 6-well plates (2 mL final volume of media each well), 1 μg plasmid DNA and 3 μL TransIT-LT1 transfection reagent (Mirus, Cat#2300) were diluted in 100 μL Opti-MEM Reduced-Serum Medium (Gibco, cat#31985062) to make the transfection mixture. The mixture was added to media in each well immediately after seeding cells ($0.5-1 \times 10^5$ cells). 24 hr after transfection, the cells were fixed at 60–70% confluency. The amount of each reagent was scaled up or down when transfecting different volume of cells.

Synthesized RNA oligos (GU-rich oligos: Cy5-mGmUmG-mUmGmUmGmUmGmUmGmU; polyA oligos: Cy5-mAmAmA-mAmAmAmAmAmAmAmAmAmA) were custom-ordered from MilliporeSigma and were resuspended as 100 μM stock solutions in RNase-free water. The RNA oligos are labeled by Cyanine5 (Cy5) at their 5' ends and have methylation at each nucleotide to prevent degradation. The cells were transfected with plasmids encoding KIF1C(ST) or KIF1C(IDR) so that KIF1C condensates were already present in cells and then 24 h later, were transfected with the synthesized RNA oligos (MilliporeSigma). The oligos at various concentrations and 4 μL Lipofectamine RNAiMAX transfection reagent (Invitrogen, Cat# 13778100) were separately diluted in 500 μL Opti-MEM (Gibco, Cat#31985062) and incubated for 5 min at room temperature. Then the two tubes were mixed to make a final mixture of 1 mL and incubated for 15 min at room temperature. 1 mL culture media was removed from each well and replaced with 1 mL transfection mixture. Cells were returned to the incubator for up to 4 hr before fixation and further processing.

## Immunofluorescence (IF)

For paraformaldehyde fixation, the cells were rinsed with PBS and fixed in 3.7% (vol/vol) paraformaldehyde (ThermoFisher Scientific) in PBS for 10 min at room temperature. Fixed cells were permeabilized in 0.2% Triton X-100 in PBS for 5 min. For methanol fixation, the cells were rinsed with PBS, then fixed and permeabilized at the same time in pre-chilled methanol for 8 min at −20 °C. After fixation and permeabilization, the cells were blocked with 0.2% fish skin gelatin in PBS for 5 min. Primary antibodies

were applied in 0.2% fish skin gelatin in PBS overnight at 4 °C in a custom humidified chamber. Following 3 washes with 0.2% fish skin gelatin in PBS, secondary antibodies were applied in 0.2% fish skin gelatin in PBS for 1 hr at room temperature in the dark. Nuclei were stained with 10.9 mM 40,6-diamidino-2-phenylindole (DAPI) and the coverslips were mounted using Prolong Gold (Invitrogen). Images were acquired on an inverted epifluorescence microscope (Nikon TE2000E) with a 40×, 0.75 NA, a 60×, 1.40 NA oil-immersion, or a 100×, 1.40 NA objective and a CoolSnap HQ camera (Photometrics). For immunofluorescence of endogenous KIF1C (Rabbit anti-KIF1C, Abcam # ab72238), cells were fixed with paraformaldehyde, and 0.02% SDS was added to the permeabilization buffer (0.2% Triton X-100 in PBS).

## Live-cell imaging

For the cytoplasm dilution assay, cells were seeded in 35 mm glass-bottom dishes (Matek, Cat# P35G-1.5-14-C). 16–24 h post-transfection, the cells were washed with and then incubated in Leibovitz's L-15 medium (Gibco) and imaged at 37 °C in a temperature-controlled and humidified stage-top chamber (Tokai Hit) on a Nikon X1 Yokogawa Spinning Disk Confocal microscope with a 60×, 1.49 NA oil-immersion objective, and a Andor DU-888 camera. Image acquisition was controlled with Elements software (Nikon). To minimize stage drifting during media exchange, a syringe was attached to the microscope stage using modeling clay to keep it static relative to the 35 mm glass-bottom dish. The syringe was used to remove media from the dish, and then new media was added by gentle pipetting. During live-cell imaging, cells were imaged in L-15 medium for more than 1 min to let the imaging process stabilize, washed once with hypotonic medium (Leibovitz's L-15 medium diluted by 1:4 with sterile water), and then incubated in the hypotonic medium to observe the change of LLPS. The dish was then washed once with the isotonic medium (Leibovitz's L-15 medium), and incubated in the isotonic medium to observe recovery.

For fluorescence recovery after photobleaching (FRAP) experiments, cells in glass-bottom dishes (Matek, Cat# P35G-1.5-14-C) were observed on a Nikon A1R line scan confocal microscope. Prebleach images were acquired, followed by photobleaching at 100% laser power at 488 nm and 80% laser power at 561 nm for 1 s, and then fluorescence recovery images were collected over time (1 frame every 3 s). Afterwards, the images were corrected for photobleaching in Fiji (https://imagej.net/software/fiji/). Fluorescence intensity in each frame was quantified in Fiji and analyzed using R scripts. The half-time of fluorescence recovery was calculated by fitting the recovery curve after the point of bleaching to an exponential curve:

$$y = A\left(1 - e^{-bx}\right),$$

where $y$ is normalized intensity, $x$ is time after bleaching. Once parameters $A$ and $b$ were calculated in R, the time constant was defined as $\tau_{1/2} = 0.693/b$. The mobile fraction was calculated by averaging the normalized intensity after the point where the recovery reaches its plateau.

For microinjection of RNase A, COS-7 cells expressing KIF1C(ST)-mNG plasmid were washed with Leibovitz's L-15 medium and then incubated in Leibovitz's L-15 medium. The dish was mounted on the stage of Nikon Eclipse (TE2000-U)

microscope and images were obtained with a CoolSNAP ES2 camera at room temperature. The microinjection was performed with glass capillaries (Eppendrof Femtotips, Cat# 930000035) loaded with 5 mg/mL of RNase A (Roche, Cat# 10109142001) mixed with Dextran-TAMRA in PBS. Before mixing, the RNase and Dextran-TAMRA aliquots were spun at 4 °C for 10 min to remove any precipitates. For control experiments, only Dextran-TAMRA in PBS was loaded in glass capillaries. The micromanipulator (Eppendorf InjectMan NI 2) was positioned and microinjection was performed using microinjector (Eppendorf FemtoJet) with a 40x dry objective lens to facilitate immediate visualization and image acquisition before and after microinjection. 40 hPa of compensation pressure (Pc) and 95 hPa injection pressure (Pi) for 1.0 second injection time (Ti) was used for each microinjection.

## Protein purification

KIF1C fragments tagged with 6xHis, SUMO and mNG in the pMW96 vector were transformed into Rosetta2 cells. Single colonies were cultured overnight, reinoculated the next day to grow to log phase, and protein expression was induced by adding 1 mM isopropyl β-D-thiogalactoside (IPTG) (Invitrogen, Cat# 15529019). After 16 h, the cells were collected by centrifugation at 20,000 × g for 3 min. Bacterial pellets were resuspended in high-salt lysis buffer (500 mM NaCl, 1% Triton X-100, 50 mM Tris pH 8.0, 10 mM MgCl₂, 0.1 mg/mL lysozyme, 1x Benzonase nuclease) by thorough vortexing. After incubation at 37 °C for 30 min, the mixture was centrifuged at 20,000 × g for 15 min at 4 °C. The soluble fraction (supernatant) was collected and mixed with Ni-NTA resin at 4 °C overnight with slow rotation. The protein-bound resin was washed 3X with wash buffer (500 mM NaCl, 1% Triton X-100, 50 mM Tris pH 8.0, 10 mM MgCl₂, 25 mM Imidazole, 1 mM DTT) and 6xHis-SUMO-TEV-KIF1C-mNG protein was eluted by elution buffer (500 mM NaCl, 1% Triton X-100, 50 mM Tris pH 8.0, 10 mM MgCl₂, 250 mM Imidazole, 1 mM DTT). Imidazole was removed by 3 cycles of concentration and dilution using Vivaspin (Millipore Sigma, Cat# GE28-9322-25). TEV enzyme was added at about 1/10 of the target protein mass and incubated overnight. The reaction mix was then incubated with Ni-NTA resin to retain the cleaved His-SUMO tag. KIF1C-mNG protein was collected in the flow-through and concentrated with Vivaspin. The protein solution was brought to 10% glycerol, aliquoted, and snap frozen in −80 °C freezer. The purity and concentration of the purified proteins were analyzed by SDS-PAGE and SimpleBlue SafeStain (Invitrogen, Cat#LC6060).

## Evaluation of liquid–liquid phase separation (LLPS) in vitro

#1.5 coverslips (Thermo Fisher Scientific) were cleaned by sonicating in 200 mM KOH (Sigma, 221473-500G) at 40 °C for 20 min. The cleaned coverslips were then incubated in aminosilane (Sigma, 440140) overnight and PEG salinization was applied the next day with mPEG-SVA (Laysan Bio, MPEG-SVA-2000-1g). To make flow chambers, the clean, PEGylated coverslips were attached to a clean glass slide (Thermo Fisher Scientific) with two thin strips of parafilm. The glass slide was briefly placed on a hotplate to slightly melt the parafilm strips, creating a seal between the coverslip and the slide. At room temperature, purified protein in

high salt buffer (500 mM NaCl, 1% Triton X-100, 50 mM Tris pH 8.0, 10 mM MgCl$_2$, 1 mM DTT) was mixed with low-salt buffer (0 mM NaCl, 1% Triton X-100, 50 mM Tris pH 8.0, 10 mM MgCl$_2$, 1 mM DTT) to reduce NaCl concentration to desired level (e.g., 100 mM NaCl) and initiate a LLPS reaction. The empty flow chamber was rinsed using salt buffer (the same NaCl concentration as in the protein mix). Then the LLPS reaction was added into the flow chamber. Images were acquired on an inverted epifluorescence microscope (Nikon TE2000E) with a 100x, 1.40 NA objective and a CoolSnap HQ camera (Photometrics) at room temperature.

## Generation of KIF1C-KO cells by CRISPR/Cas9-mediated genome editing

Knockout of KIF1C from the genome of hTERT-RPE1 cells was performed as described previously (Ran et al, 2013). The 20-nt sgRNA sequences were designed by the CRISPR guide RNA design tool in Benchling (https://benchling.com). The oligo pairs [forward strand: 5'-aaacCGGTGAAAGTGGCAGTGAGGc-3'; reverse strand: 5'-caccgCCTCACTGCCACTTTCACCG-3'] were synthesized (IDT), annealed, and ligated into plasmid eSpCas9(1.1). The product plasmid was transfected into wild-type hTERT-RPE1 cells using TransIT-LT1 transfection reagent (Mirus, Cat#2300). Forty-eight hours post-transfection, the cells were selected in 800 µg/mL Geneticin (Gibco) until cells in the untransfected control group completely died. Single cells were then isolated through flow cytometry and sorted into 96-well plates. Colonies were expanded and genomic DNA was extracted. The KIF1C sequence close to the sgRNA target site was amplified by PCR and sequenced. KIF1C-KO cell lines identified by DNA sequencing were verified for loss of KIF1C expression by Western Blot using Rb anti-KIF1C antibody.

## Generation of stably transfected cells using KIF1C-KO cells

KIF1C KO hTERT-RPE1 cells were transfected with pN1-HsKIF1C(FL)-mNG or pN1-HsKIF1C(ΔIDR)-mNG plasmids (see Key Resources Table) using TransIT-LT1 transfection reagent (Mirus, Cat#2300). Forty-eight hours post-transfection, the cells were selected in 800 µg/mL Geneticin (Gibco) until cells in the untransfected control group completely died. The surviving cells were expanded and cells with low levels of mNG expression were selected by flow cytometry and further expanded as a mixed culture.

## Estimation of endogenous KIF1C concentration by western blot (WB)

hTERT-RPE1 parental cells were trypsinized and harvested by centrifugation at 5000 × g at 4 °C for 5 min, washed with cold 1X PBS, and resuspended in cold lysis buffer [25 mM HEPES/KOH, 115 mM potassium acetate, 5 mM sodium acetate, 5 mM MgCl$_2$, 0.5 mM EGTA, and 1% (vol/vol) Triton X-100, pH 7.4] with 1 mM ATP, 1 mM phenylmethylsulfonyl fluoride (PMSF), and 1% (vol/vol) protease inhibitor cocktail (P8340, Sigma-Aldrich). Lysates were clarified by centrifugation at 20,000 × g at 4 °C for 10 min and the supernatants were mixed with 5X Laemmli buffer and 1: 50 1 mM DTT. The samples were boiled at 95 °C for 5 min, cooled to room temperature and briefly centrifuged before loading into SDS-PAGE gels and immunoblotted using Rb anti-KIF1C antibody (Abcam, Cat#ab72238). Purified

KIF1C(IDR)-mNG was used to generate a protein standard. The mass of KIF1C was calculated by fitting the intensity of protein band to the KIF1C(IDR)-mNG standard curve. The volume of cytoplasm was estimated by multiplying the number of cells loaded and the average volume of each cell. For example, $1.35 \times 10^6$ hTERT-RPE1 cells × 2416 ± 263 µm$^3$/cell [mean ± SD, (Pollreisz et al, 2020)] = 3.3 µL of cytoplasm loaded into each lane. The endogenous KIF1C concentration was calculated by dividing the mass of KIF1C by the cytoplasmic volume and protein molecular weight, and was averaged across 3 experiments. Note that due to loss of protein during the process of making lysates, the measured value is likely to be an underestimate.

## Single-molecule fluorescence in situ hybridization (smFISH)

Single-molecule fluorescence in situ hybridization and the sequences of probes against *RAB13*, *NET1*, and *CMA1* mRNAs were based on previous literature (Tsanov et al, 2016; Haimovich and Gerst, 2018; Pichon et al, 2021; Calvo et al, 2021). The primary and secondary probes were annealed and diluted in hybridization buffer (10% Formamide, 1 mg/mL *E. coli* tRNA, 10% Dextran Sulfate, 0.2 mg/mL BSA, 2x Saline-sodium citrate (SSC) buffer, 2 mM Vanadyl ribonucleoside complex). Cells were washed 3X with PBSM (PBS with 5 mM MgCl$_2$), fixed with 3.7% (vol/vol) paraformaldehyde in PBSM for 10 min at room temperature, washed 2X with PBSM, and permeabilized with permeabilization buffer (0.1% Triton X-100 in PBSM) for 10 min. Cells were then washed 2X and incubated with pre-hybridization buffer (10% Formamide in 2x SSC) for 30 min at RT, and the incubated with annealed probes in hybridization buffer overnight at 37 °C in a custom-made humidified chamber. The next day, the cells were washed twice with pre-hybridization buffer at 37 °C in humidity chamber, and three times with 2x SSC at room temperature. 0.5 µg/mL DAPI staining solution was applied for 1 min at room temperature. The cells were washed three times and finally mounted with ProLong Gold (Invitrogen). Images were acquired on an inverted epifluorescence microscope (Nikon TE2000E) with a 40×, 0.75 NA, a 60×, 1.40 NA oil-immersion, or a 100×, 1.40 NA objective and a CoolSnap HQ camera (Photometrics).

For co-staining of mRNAs by smFISH and KIF1C protein by immunofluorescence (Abcam # ab72238), the IF staining of KIF1C was performed first using RNase-free conditions and reagents. After the secondary antibody incubation, a postfixation step was applied by incubating the sample in 4% PFA for 10 min at room temperature. Then the sample was washed by 2x SSC for several times and incubated in pre-hybridization buffer and the smFISH protocol above was carried out.

## Image analysis

Image analysis was carried out using Fiji/ImageJ (https://fiji.sc). Experimenters were not blinded to group assignment and outcome assessment.

For quantification of KIF1C and KIF16B puncta sizes (Fig. 1C), puncta that were aggregated in cell protrusions were excluded from quantification as they could not be individually defined.

For analysis of KIF1C enrichment in puncta (Fig. 2E), measurements were made on a single cell basis of the mean fluorescence intensity value of all puncta ($I_{puncta}$) and compared to

the mean fluorescence intensity of an area adjacent to the puncta ($I_{\text{diffusive}}$). The background intensity ($I_{\text{background}}$) was measured from neighboring untransfected cells and the final ratio was calculated by the following equation:

$$\text{Intensity ratio} = \frac{I_{\text{puncta}} - I_{\text{background}}}{I_{\text{diffusive}} - I_{\text{background}}}\ .$$

The calculated intensity ratios were plotted as a violin plot in R where each point represents measurement from one cell.

Colocalization between KIF1C puncta and *RAB13* mRNA (Figs. 5C,D and 8F) was measured by drawing a segmented line across KIF1C puncta and counting the number of *RAB13* mRNA puncta that colocalize with KIF1C puncta along the line. A KIF1C punctum was defined as a fluorescent peak along the line where the intensity value is at least 1.5-fold the average intensity along the line. A colocalizing *RAB13* mRNA peaks was defined as a fluorescence peak within 0.4 μm distance (~3 pixels) of the KIF1C punctum.

When quantifying puncta size of different IDR variants of KIF1C in each cell (Fig. 6F), a weighted mean was calculated by:

$$\bar{d} = \frac{\sum_{i=1}^{n} A_i d_i}{\sum_{i=1}^{n} A_i}\ ,$$

where $d_i$ is the diameter of individual puncta in a cell, $A_i$ is the corresponding area of puncta. When there are a couple of dominating puncta in a cell that contain most of dense phase KIF1C, this weighted mean (shown above) can represent this situation better than an ordinary arithmetic mean.

When quantifying the level of RNA enrichment in CC4 + IDR-mNG condensates in vitro (Fig. 7F), the CC4 + IDR-mNG channel and the RNA channel from $25 \times 25$ μm field of views were filtered by Gaussian Blur (sigma = 1), then segmented using auto local threshold (Bernsen, radius = 70–100). Global threshold (Yen) was applied when the background noise is too high to allow faithful auto local thresholding. RNA enrichment level is represented by an RNA colocalization ratio, which is the overlapping area between the CC4 + IDR-mNG condensates and RNA condensates divided by the total area of CC4 + IDR-mNG condensates.

$$\text{RNA colocalization ratio} = \frac{\text{overlapping area with RNA condensates}}{\text{total area of CC4 + IDR condensate}}\ .$$

Note that although the intensity of the RNA channel varies due to different laser power used for different RNA concentrations, this RNA colocalization ratio is not sensitive to the absolute intensity values.

## Data availability

KIF1C bioID data plotted in Appendix Fig. S6 are from (Kendrick et al, 2019). An interactive plot is accessible here (https://cdb-rshiny.med.umich.edu/Geng-KIF1C_bioID_visualization/). The code and data for making the interactive plot are available on GitHub (https://github.com/Archie-G/KIF1C_bioID_visualization.git). The source data for Fig. 3E is available on Bioimage Archive (accession number S-BSST1398).

The source data of this paper are collected in the following database record: biostudies:S-SCDT-10_1038-S44318-024-00147-9.

## Peer review information

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

## Acknowledgements

We thank the members of the Verhey, Cianfrocco, DeSantis, Ohi and Sept labs (University of Michigan) for ideas, discussion, feedback, and support. We thank Stravoula Mili and Konstadinos Moissoglu (National Cancer Institute, NIH) for discussion, feedback and protocols. We are grateful to our colleagues at the University of Michigan: Craig Johnson for help with making the RShiny app publicly available, Zesen Lin and Rami Khoriaty for help with CRIPSR-Cas9 protocols for generating KIF1C KO cells, Amanda Erwin and Shyamal Mosalaganti for help with experiments using fluorescently-labeled synthetic RNA oligos, John Gillies and Morgan DeSantis for help with PEG salinization of coverslips, and Ye Yuan and Swathi Yadlapalli for help with smFISH. We are grateful to Eric Rentchler in the University of Michigan Biomedical Research Core facilities for training and guidance. This work was supported by funding from the National Institutes of Health to KJV (R35GM131744).

## Author contributions

**Qi Geng**: Conceptualization; Data curation; Software; Formal analysis; Validation; Investigation; Visualization; Methodology; Writing—original draft; Writing—review and editing. **Jakia Jannat Keya**: Formal analysis; Investigation. **Takashi Hotta**: Investigation. **Kristen J Verhey**: Conceptualization; Resources; Supervision; Funding acquisition; Writing—original draft; Project administration; Writing—review and editing.

Source data underlying figure panels in this paper may have individual authorship assigned. Where available, figure panel/source data authorship is listed in the following database record: biostudies:S-SCDT-10_1038-S44318-024-00147-9.

## Disclosure and competing interests statement

The authors declare no competing interests.

