## [Peer Review File · The EMBO Journal]

The kinesin-3 KIF1C undergoes liquid-liquid phase separation for accumulation of specific transcripts at the cell periphery

Qi Geng, Jakia Keya, Takashi Hotta, and Kristen Verhey

Corresponding author(s): Kristen Verhey (kjverhey@umich.edu)

Review Timeline:

Transfer from Review Commons:	19th Mar 24
Editorial Decision:	22nd Apr 24
Revision Received:	30th Apr 24
Accepted:	17th May 24

Review
COMMONS

Editor: William Teale

Transaction Report: This manuscript was transferred to The EMBO JOURNAL following peer review at Review Commons.

Review #1

1. Evidence, reproducibility and clarity:

Evidence, reproducibility and clarity (Required)

Summary

Geng et al. explore the molecular mechanisms underlying the role of KIF1C in RNA transport, focusing on how it interacts with RNA. KIF1C is shown to form dynamic puncta when overexpressed in COS-7 cells that do not appear to colocalise with organelle markers. An IDR in the tail of the kinesin is necessary and sufficient for the formation of these structures and FRAP experiments show that they can exchange their contents with proteins in the cytosol and that their formation can be reversibly modulated by hypotonic shock, consistent with LLPS. In vitro, the IDR and flanking regions can undergo phase separation at physiologically relevant concentrations and salt conditions. In cells, KIF1C puncta enrich for RNAs and support their transport, and depletion of RNA modulates KIF1C LLPS properties. A model is proposed whereby KIF1C mediated RNA transport to the cell periphery promotes the formation of a protein-RNA condensate that may act to fine tune local RNA activity.

Major comments

In general, the claims made here are well-supported by the data. However, I think that some exploration of the extent of LLPS at different KIF1C expression levels in cells is important but missing. The authors carefully estimate the endogenous concentration of KIF1C in COS-7 cells (at around 25 nM), but it isn't clear how this compares to that observed in transient transfection experiments. Although this is partly addressed in the in vitro assays, I am still left with some questions over the extent of this phenomenon in a cellular context. Can the authors provide some experimental evidence to support the proposition that LLPS occurs (perhaps in a more localised fashion?, as Fig.9) at lower KIF1C expression levels? One way to address this might be a GFP-knock-in (although how feasible this is may depend on the genomic context), alternatively, the authors could generate cell lines that express KIF1C-GFP from a very weak promoter, demonstrate LLPS using their established assays, show that this is comparable to endogenous expression.

Minor comments

Lines 107-109 and Figure 1B on localisation of other kinesin-3s. The authors state that they localise to certain organelle but don't show co-staining for those organelles.

Lines 172-183 and Figure 3. Evidence is provided through FRAP experiments that KIF1C puncta exchange with the cytosolic pool. However, the extent of recovery appears to saturate at <40%. Does this suggest the existence of an immobile pool of KIF1C within these structures?

Line 238 - Fig. S5C is cited as data on endogenous concentration of KIF1C - this should be Fig. S6C.

Line 331-332 - I did not fully follow the logical here the RNase A injection experiment supports the idea that KIF1C interaction with RNA is sequence selective. Could the authors expand on this.

2. Significance:

Significance (Required)

This study introduces a new and exciting concept to motor protein biology: that some cytoskeletal motors and motor-cargo complexes can undergo phase separation, and that this is important for their function. The experiments are logical, progressive, and form a clear and compelling case. The main limitation is that demonstration of LLPS in cells is limited to over-expressed protein. Some exploration/demonstration of LLPS properties of KIF1C in cells at near to endogenous expression levels would enhance the study.

The work should be of interest to a broad range of readers, from the cytoskeletal motor community, those interested in mRNA regulation, as well as scientists studying phase separation more generally.

3. How much time do you estimate the authors will need to complete the suggested revisions:

Estimated time to Complete Revisions (Required)

(Decision Recommendation)

Between 1 and 3 months

No

Review #2

1. Evidence, reproducibility and clarity:

Evidence, reproducibility and clarity (Required)

This paper investigates mRNA transport by the kinesin Kif1C and tests the hypothesis that liquid condensation of the disordered C terminal region is important for mRNA recruitment. It is based on prior work from other labs showing that Kif1C recruits and transports a set of mRNAs to the periphery of cells. The mechanism of the Kif1C-mRNA interaction was not investigated in the prior work, so the proposal that a liquid condensate is involved is novel. It is also topical, since there is intense current interest in transport and regulation of mRNAs by condensate-mediated mechanisms. The most useful part of this paper to the field may be the identification of IDR2 as required for mRNA binding in Fig 7

A major concern is reliance on expression of tagged Kif1C in Cos cells in several figures. The expression level in these experiments probably far exceeds normal, though this comparison is not reported. It is possibly justified to use over-expression to reveal a condensate mechanism, but it is concerning and the authors need to strongly qualify their conclusions. One way to moderate this concern would be to examine condensation as a function of expression level.

Another significant concern is that the biochemical reconstitution figure tests protein alone, not protein + RNA. Disordered RNA binding proteins usually phase separate

better in the presence of RNA. The best reconstitution papers evaluate specificity of RNA recruitment to condensates. Specificity testing in a reconstituted system may not be required for a first paper, but testing the effect of some kind of RNA seems important.

A final concern is that the specificity of mRNA recruitment to Kif1C puncta in cells is not critically evaluated. Among endogenous mRNAs, only one (Rab13) is tested. The paper would be stronger with a second positive mRNA and a negative control mRNA.

2. Significance:

Significance (Required)

The mechanism of the KifC1-mRNA interaction was not investigated in the prior work, so the proposal that a liquid condensate is involved is novel. It is also topical, since there is intense current interest in transport and regulation of mRNAs by condensate-mediated mechanisms. The most useful part of this paper to the field may be the identification of IDR2 as required for mRNA binding.

3. How much time do you estimate the authors will need to complete the suggested revisions:

Estimated time to Complete Revisions (Required)

(Decision Recommendation)

Cannot tell / Not applicable

Yes

Review #3

1. Evidence, reproducibility and clarity:

Evidence, reproducibility and clarity (Required)

KIF1C is a member of the kinesin-3 family, which is responsible for fast organelle transport in cells. The cargos of KIF1C are diverse, such as Golgi apparatus, Rab6 vesicles, exon junction complex (EJC), integrins, and RNA. Mutations in the KIF1C coding sequence leads to neurodegenerative diseases, such as hereditary spastic paraparesis (HSP). In addition, as an RNA transporter, KIF1C transports various types of mRNAs (e. g., APC-dependent mRNAs, KIF1C's own mRNA) along the microtubules and clusters them to cytoplasmic protrusions to fulfill certain biological functions.

In the current manuscript, Gen et.al., investigated the intracellular behaviors of the kinesin-3 member KIF1C. The study revealed that the KIF1C can form dynamic condensates both in cells and in vitro via an unstructured domain within the tail of the motor. KIF1C was found to also interact with synthesized RNA and other RNA granules in cells. In addition, the authors also show the KIF1C participates intracellular transport of endogenous mRNA, Rab13mRNA, identified a 47aa fragment in the KIF1C's IDR is critical for the KIF1C- Rab13mRNA interaction. Finally, as well as other prion-like proteins, the PPLS of KIF1C is buffered by the non-specific RNA pool in the cytoplasm.

In summary, this is an interesting work in the field, and reveals novel results about the mechanisms of motor protein transport that will be broadly interesting. The assays are generally well performed, and the results and discussion are well described, but some descriptions in the article should be more rigorous and objective. The article is very long, and I think it would benefit from streamlining and reducing the number of figures to make it more accessible for non-specialists in the field.

Here are some concerns:

****Major:****

1. Fig. 1A shows the domain organization of all kinesin-3 members, but Figure 1B only represents KIF1B β , KIF13B and KIF16B as controls. Generally, the KIF1B α has the highest sequence similarity with KIF1C in kinesin-3 family (very high sequence similarity before aa 992 in KIF1C, which locates in IDR, probably contains IDR2a from Fig. S10A). In addition, both KIF1C and KIF1B α contain a PLD from the prediction in this paper (Figure S2C). Although the authors show the phenotype of KIF1B α in the Fig. S9, it might be better to put some descriptions up front, as readers may consider why the authors did not use KIF1B α as a control. Actually, I kept thinking about this concern before I got to the discussion.

2. It would be better if the authors can combine the Fig. 2B and 2C, since the article did not mention Fig. 2B at all. In addition, Fig. S3 does not help this article too much. Probably it would be better if the authors could take the Δ IDR-mNG data from the Fig. S3 and put into the Fig. 2. as a negative control, especially for Fig. 2D and 2F. As for whether the phenotype of the Δ IDR-mNG construct is "similar to a constitutively active KIF1C construct containing only the motor domain (amino acids 1-348) (Fig. S3 C)", I do not think it is important here, since in this part, the authors are aiming to confirm the IDR is critical for KIF1C phase separation.

3. The description "the condensate properties can be modulated by adjacent coiled-coil segments" in the abstract and the sentence "However, the coiled-coil segments in the stalk domain appear to facilitate puncta formation as the addition of increasing amounts of coiled coil resulted in increased KIF1C enrichment in puncta as compared to the IDR alone" in the article are not accurate, since there is no direct evidence in this manuscript that shows that. In Fig. 2D, as well as Fig. 6A and Fig. S7, it is manifest at a glance there are lots of IDR-mNG localized in nucleus, which decreases the concentration of this construct in cytoplasm which in turn may lower its capability to form puncta. This is important, as the results in Fig.4 show that the concentration of protein directly affects the formation of phase separated puncta in cells. From my view, the words "modulate", "tune" ... usually describe active processes, and these words may be confusing unless there are enough evidence support direct regulation. But the data presented in this article suggests to us that it is likely a passive process, such as the coiled coil region preventing the CC4-IDR construct from entering the nucleus (Fig. 2D, Fig. 6A and Fig. S7). Moreover, CC4 does affect the critical concentration of IDR in vitro (Fig. 5E), but that could be attributed to the coiled coil domain increasing its solubility. I like the word "influence" used in a subtitle in the discussion portion.

In addition, the in vitro study of this paper in Fig. 5 did not show any significant difference of the puncta formation between IDR-mNG and CC4 - IDR-mNG (Diameter: $0.43 \pm 0.22 \mu\text{m}$ (mean \pm STD) for IDR-mNG vs $0.48 \pm 0.27 \mu\text{m}$ (mean \pm STD) for CC4-IDR-mNG. Roundness: No value was show in the article). So, a stricter assay or a more accurate

description is required here to avoid any misleading to the readers.

The description for Fig. 5 "At 2 μM protein concentration and 100 mM NaCl, the KIF1C(IDR) droplets were smaller [diameter $0.43 \pm 0.22 \mu\text{m}$ (mean \pm STD)] than KIF1C(CC4+IDR) droplets [$0.48 \pm 0.27 \mu\text{m}$ (mean \pm STD)] (Fig. 5 C)" does not appear accurate as well, since there is no significant difference between the value $0.43 \pm 0.22 \mu\text{m}$ and the value $0.48 \pm 0.27 \mu\text{m}$, so it should not be described as "smaller". In addition, the article mentioned that "The KIF1C(IDR) puncta were also less round than those of KIF1C(CC4+IDR) (Fig. 5 C)", but there is no corresponding value from the quantification show the KIF1C(IDR) is less round.

4. The description in sentence "We thus tested whether ... LLPS is mutually exclusive (Fig. S5 A)" may not be accurate. Results in Fig. S5 only show there is no direct interaction between KIF1C and CLIP-170 or these two proteins do not colocalize. The words "mutually exclusive" means two proteins competent each other in the same location from my understanding.

In addition, is it necessary to put Fig. S5 into this article? Since from my side, it does not help too much for the whole story. In cells, kinesin motors are autoinhibited in the cytoplasm. For this KIF1C, most of motors appear autoinhibited as well, even when the authors removed the IDR based on Fig. S3C (ΔIDR -mNG vs. MD-mNG). In this case, it is hard to investigate the potential interaction between the KIF1C (or its ΔIDR mutant) with the microtubules or with the tubulin due to the autoinhibition of constructs used in Fig. S5. It would be better to use other active versions of KIF1C, such as ΔP (Soppina et. al., PNAS, 2014) or other mutants (Ren et. al., PNAS, 2018; Wang et. al., Nat. Commun., 2022) if the authors want to show this part in the article.

5. The conclusion "This result suggests that the IDR- driven LLPS of KIF1C does not depend on mRNA incorporation, but is strongly affected by it" may not be accurate, there is no direct evidence that shows that mRNA, at least Rab13mRNA incorporation strongly affects the IDR- driven LLPS of KIF1C. Perhaps a knock out of Rab13mRNA would alter the formation of condensates, which would support a direct effect on LLPS.

In addition, the sentence "These results also show that the LLPS is resistant to truncations of large portions of IDR" may not accurate, from my view, except IDR2a, the rest of the IDR may not participate or contribute too much to the formation of puncta, but that doesn't mean LLPS is resistant to the truncation of these portions in IDR, these are different logics. The quantification from Fig. 7E also show there is no significant difference between the ST and truncations except ΔIDR2 and ΔIDR2a in statistics, such as ST (21.8 ± 12.0 puncta per cell, $2.06 \pm 0.83 \mu\text{m}$ diameter), ΔPLD (20.1 ± 13.7 puncta per cell, 1.62 ± 0.94

μm diameter), ΔIDR1 (23.1 \pm 14.3 puncta per cell, 2.13 \pm 1.04 μm diameter), ΔIDR3 (18.4 \pm 8.1 puncta per cell, 1.71 \pm 0.98 μm diameter).

6. I am not sure I agree with the author's interpretation of their FRAP data in Fig. 3. It appears to me that there is a large immobile population of molecules, as the bleached areas recover less than 50% of their initial intensity. However, the authors conclude that there is rapid exchange of molecules in the puncta. The authors need to further analyze and discuss both the exchange rate of the population of molecules that exchange, but also the fraction of apparently immobile molecules that do not recover in their experiments. These data appear to suggest that a large percentage of the molecules in the KIF1C puncta in fact do not exchange with the cytoplasm and undermine their argument for a liquid-like phase of the puncta.

****Minor:****

1. As mentioned above, Fig. 2 F needs a negative control, since the values of FL and IDR are lower than other constructs, maybe use the Δ IDR-mNG protein is better. In addition, from my view, the lower value of IDR construct does not represent this construct has lower capability to form puncta, but more likely because most of this protein localizes in nucleus, thus dramatically lowering the cytoplasmic concentration.
2. Fig. 6A probably need a negative control as well, maybe use the same construct ΔIDR in Fig. S7 is better.
3. Although I guess the reason for using hTERT-RPE1 cells in Rab13mRNA rescue assay (Fig. 6D-G) probably is easier to get KIF1C knock out cells (if I am correct), it would be better if there is a brief introduction for the reason to use hTERT-RPE1 here, since all previous assay in the article used COS-7 cells.
4. Is there any specific reason to use the construct ST in Fig. 7? Since in Fig. 6, the authors used FL-length KIF1C, if the authors want to avoid any effects caused by motor domain, the construct CC4-IRD also could be a simpler candidate.
5. This article is a great case for motor-cargo interaction, since the RNA binding site of KIF1C is within its tail domain. This left me curious about if the interaction between the KIF1C and the membrane-less RNA granule is sufficient to release the KIF1C motor from autoinhibition? I guess the binding of RNA is not enough to release the KIF1C from autoinhibition. From Fig. S3C and Fig. 6D, seems the motor still in autoinhibition, even remove the Rab13mRNA binding region.
6. There are some grammar mistakes, e.g., There should be a "is" between "IDR" and "critical" in the title "A subregion of the KIF1C IDR critical for enrichment of Rab13mRNA in condensates".
7. There should be a definition for the full names of the abbreviate "RBD" mentioned in the article although the readers may guess that is an RNA binding domain, if

possible, it would be better but not necessary if the authors could show the residues or the region in IDR.

8. In the results (line 126), the authors refer to the KIF1C IDR without first defining this region in the introduction. I would re-word this sentence for clarity by first defining what an IDR is and how it's assessed in the current study.

9. What is the significance of the roundness measurement in Fig. 5? This should be described for the reader.

10. The authors state several times that this is the first kinesin shown to undergo LLPS. However, is this true? What about the recent work showing that the yeast Tea2 kinesin undergoes LLPS with other +TIP components (Maan et al. NCB 2023).

11. The authors don't discuss KIF5A, but their analysis reveals it also contains a low complexity region that may undergo LLPS (Fig. S2D). This would fit with recent reports that KIF5A tends to oligomerize more than other KIF5 isoforms, and that mutations in KIF5A that impact the tail domain may lead to aberrant oligomerization. I feel that it would be useful to the field for the authors to discuss these results in light of their own.

2. Significance:

Significance (Required)

The study is novel and interesting and will be impactful for the cytoskeletal and RNA biology communities. The experiments are of high quality and controls are appropriate. The finding that motor proteins can participate in LLPS will be of high interest for a variety of fields and provides a very interesting advance over current knowledge in the field.

3. How much time do you estimate the authors will need to complete the suggested revisions:

Estimated time to Complete Revisions (Required)

(Decision Recommendation)

Between 1 and 3 months

4. *Review Commons* values the work of reviewers and encourages them to get credit for their work. Select 'Yes' below to register your reviewing activity at Web of Science Reviewer Recognition Service (formerly Publons); note that

the content of your review will not be visible on Web of Science.

Yes

Full Revision

Manuscript number: RC-2023-02220

Corresponding author(s): Kristen Verhey

[Please use this template only if the submitted manuscript should be considered by the affiliate journal as a full revision in response to the points raised by the reviewers.]

1. General Statements [optional]

This section is optional. Insert here any general statements you wish to make about the goal of the study or about the reviews.

We thank the reviewers for their time and effort in reviewing our manuscript. We generated new tools (e.g. smFISH probes) and cell lines and changed the text for increased accuracy and clarity. We believe that the changes have improved the manuscript. Below, we respond to each of the reviewers concerns with our responses indicated in blue text. Changes within the manuscript are also in blue text.

This section is mandatory. Please insert a point-by-point reply describing the revisions that were already carried out and included in the transferred manuscript.

Reviewer #1 (Evidence, reproducibility and clarity (Required)):

Summary

Geng et al. explore the molecular mechanisms underlying the role of KIF1C in RNA transport, focusing on how it interacts with RNA. KIF1C is shown to form dynamic puncta when overexpressed in COS-7 cells that do not appear to colocalise with organelle markers. An IDR in the tail of the kinesin is necessary and sufficient for the formation of these structures and FRAP experiments show that they can exchange their contents with proteins in the cytosol and that their formation can be reversibly modulated by hypotonic shock, consistent with LLPS. In vitro, the IDR and flanking regions can undergo phase separation at physiologically relevant concentrations and salt conditions. In cells, KIF1C puncta enrich for RNAs and support their transport, and depletion of RNA modulates KIF1C LLPS properties. A model is proposed whereby KIF1C mediated RNA transport to the cell periphery promotes the formation of a protein-RNA condensate that may act to fine tune local RNA activity.

Major comments

In general, the claims made here are well-supported by the data. However, I think that some exploration of the extent of LLPS at different KIF1C expression levels in cells is important but missing. The authors carefully estimate the endogenous concentration of KIF1C in COS-7 cells (at around 25 nM), but isn't

clear how this compares to that observed in transient transfection experiments. Although this is partly addressed in the vitro assays, I am still left with some questions over the extent of this phenomenon in a cellular context. Can the authors provide some experimental evidence to support the proposition that LLPS occurs (perhaps in a more localised fashion?, as Fig.9) at lower KIF1C expression levels? One way to address this might be a GFP-knock-in (although how feasible this is may depend on the genomic context), alternatively, the authors could generate cell lines that express KIF1C-GFP from a very weak promoter, demonstrate LLPS using their established assays, show that this is comparable to endogenous expression.

Response: We thank the reviewer for this suggestion. We have carried out additional experiments to explore the extent of KIF1C LLPS at endogenous levels. We used antibody against KIF1C to stain WT and KIF1C knockout (KO) cells. Although the antibody shows a high background of non-specific signal in the cytoplasm and nucleoplasm of both WT and KO cells, we were able to observe small puncta of KIF1C at the periphery of WT but not KO cells (new Figure 8). This finding supports our hypothesis that endogenous KIF1C undergoes LLPS upon reaching a high local concentration at the periphery of cells.

Two lines of evidence support that these puncta of endogenous KIF1C protein are RNA-containing biomolecular condensates formed by LLPS (new Figure 8). First, these small puncta of endogenous KIF1C incorporate *RAB13* mRNA, suggesting that they are RNA granules. Second, the puncta do not form in cells stably expressing KIF1C Δ IDR at near-endogenous levels.

Minor comments

Lines 107-109 and Figure 1B on localisation of other kinesin-3s. The authors state that they localise to certain organelle but don't show co-staining for those organelles.

Response: The localization of other kinesin-3s to certain organelles has been shown in the cited literature. In response to the reviewer's request, we now verify these findings by staining cells expressing the other kinesin-3s for specific organelles (new Figure S1 A).

Lines 172-183 and Figure 3. Evidence is provided through FRAP experiments that KIF1C puncta exchange with the cytosolic pool. However, the extent of recovery appears to saturate at <40%. Does this suggest the existence of an immobile pool of KIF1C within these structures?

Response: We agree that the data suggest the existence of an immobile pool of KIF1C within the condensates. We have added this information to the main text (lines 178-182). We note that these findings are consistent with recent studies demonstrating membrane-less organelles with at least partially solid-like properties, including nucleoli and stress granules as well as microtubule associated proteins (see references, reviewed in Van Treeck & Parker 2019).

Line 238 - Fig. S5C is cited as data on endogenous concentration of KIF1C - this should be Fig. S6C.

Response: Thank you. We have corrected this (now Fig S8 C).

Line 331-332 - I did not fully follow the logic here the RNase A injection experiment supports the idea that KIF1C interaction with RNA is sequence selective. Could the authors expand on this.

Response: We thank the reviewer for this comment. We have rewritten the text (lines 235-238, 246-248).

Reviewer #1 (Significance (Required)):

This study introduces a new and exciting concept to motor protein biology: that some cytoskeletal motors and motor-cargo complexes can undergo phase separation, and that this is important for their function. The experiments are logical, progressive, and form a clear and compelling case. The main limitation is that demonstration of LLPS in cells is limited to over-expressed protein. Some exploration/demonstration of LLPS properties of KIF1C in cells at near to endogenous expression levels would enhance the study.

The work should be of interest to a broad range of readers, from the cytoskeletal motor community, those interested in mRNA regulation, as well as scientists studying phase separation more generally.

Reviewer #2 (Evidence, reproducibility and clarity (Required)):

This paper investigates mRNA transport by the kinesin Kif1C and tests the hypothesis that liquid condensation of the disordered C terminal region is important for mRNA recruitment. It is based on prior work from other labs showing that Kif1C recruits and transports a set of mRNAs to the periphery of cells. The mechanism of the Kif1C-mRNA interaction was not investigated in the prior work, so the proposal that a liquid condensate is involved is novel. It is also topical, since there is intense current interest in transport and regulation of mRNAs by condensate-mediated mechanisms. The most useful part of this paper to the field may be the identification of IDR2 as required for mRNA binding in Fig 7.

Major comments

A major concern is reliance on expression of tagged Kif1C in Cos cells in several figures. The expression level in these experimental probably far exceeds normal, though this comparison is not reported. It is possibly justified to use over-expression to reveal a condensate mechanism, but it is concerning and the authors needs to strongly qualify their conclusions. One way to moderate this concern would be to examine condensation as a function of expression level.

Response: We thank the reviewer for this suggestion. We have carried out additional experiments to explore the extent of KIF1C LLPS at endogenous levels. We used antibody against KIF1C to stain WT and KIF1C knockout (KO) cells. Although the antibody shows a high background of non-specific signal in the cytoplasm and nucleoplasm of both WT and KO cells, we were able to observe small puncta of KIF1C at the periphery of WT but not KO cells (new Figure 8). This finding supports our hypothesis that endogenous KIF1C undergoes LLPS upon reaching a high local concentration at the periphery of cells.

Two lines of evidence support that these puncta of endogenous KIF1C protein are RNA-containing biomolecular condensates formed by LLPS (new Figure 8). First, these small puncta of endogenous KIF1C incorporate RAB13 mRNA, suggesting that they are RNA granules. Second, the puncta do not form in cells stably expressing KIF1C Δ IDR at near-endogenous levels.

Another significant concern is that the biochemical reconstitution figure tests protein alone, not protein + RNA. Disordered RNA binding proteins usually phase separate better in the presence of RNA. The best

Full Revision

reconstitution papers evaluate specificity of RNA recruitment to condensates. Specificity testing in a reconstituted system may not be required for a first paper, but testing the effect of some kind of RNA seems important.

Response: The purified CC4+IDR and IDR constructs form condensates at low μM concentrations and in the absence of RNA or crowding agents, thus we did not test whether they would phase separate better in the presence of RNA. In response to the reviewer's comments, we now evaluate the specificity of RNA recruitment to the KIF1C condensates. We utilized the purified CC4+IDR protein and added the same GU-rich and polyA RNAs used in cells (now Fig 4 B) at different concentrations. Interestingly, there is selective incorporation of GU-rich oligos in condensates at low RNA concentrations, incorporation of both RNAs into condensates at medium concentrations, and an inhibition of condensate formation at high RNA concentrations (new Fig 7 E,F).

A final concern is that the specificity of mRNA recruitment to Kif1C puncta in cells is not critically evaluated. Among endogenous mRNAs, only one (Rab13) is tested. The paper would be stronger with a second positive mRNA and a negative control mRNA.

Response: We have now tested whether the specificity of mRNA recruitment to KIF1C puncta applies to additional mRNAs. We carried out single-molecule FISH (smFISH) experiments for two additional mRNAs. Based on the literature showing KIF1C-dependent localization of specific RNAs, we chose *NET1* as a second positive mRNA and *CAM1* as a negative control mRNA (Pichon et al., 2021). We first show that *NET1* mRNA is mislocalized in KIF1C KO cells whereas *CAM1* mRNA is not (new Fig S7 C,D). We then rescued the KO cells with FL or ΔIDR constructs and show that the FL protein rescues *NET1* mRNA localization to the cell periphery whereas the ΔIDR construct does not (new Fig S7 E,F).

Reviewer #2 (Significance (Required)):

The mechanism of the KifC1-mRNA interaction was not investigated in the prior work, so the proposal that a liquid condensate is involved is novel. It is also topical, since there is intense current interest in transport and regulation of mRNAs by condensate-mediated mechanisms. The most useful part of this paper to the field may be the identification of IDR2 as required for mRNA binding.

Reviewer #3 (Evidence, reproducibility and clarity (Required)):

KIF1C is a member of the kinesin-3 family, which is responsible for fast organelle transport in cells. The cargos of KIF1C are diverse, such as Golgi apparatus, Rab6 vesicles, exon junction complex (EJC), integrins, and RNA. Mutations in the KIF1C coding sequence leads to neurodegenerative diseases, such as hereditary spastic paraparesis (HSP). In addition, as an RNA transporter, KIF1C transports various types of mRNAs (e. g., APC-dependent mRNAs, KIF1C's own mRNA) along the microtubules and clusters them to cytoplasmic protrusions to fulfill certain biological functions.

In the current manuscript, Gen et.al., investigated the intracellular behaviors of the kinesin-3 member KIF1C. The study revealed that the KIF1C can form dynamic condensates both in cells and in vitro via an unstructured domain within the tail of the motor. KIF1C was found to also interact with synthesized RNA

and other RNA granules in cells. In addition, the authors also show the KIF1C participates intracellular transport of endogenous mRNA, Rab13mRNA, identified a 47aa fragment in the KIF1C's IDR is critical for the KIF1C- Rab13mRNA interaction. Finally, as well as other prion-like proteins, the PPLS of KIF1C is buffered by the non-specific RNA pool in the cytoplasm.

In summary, this is an interesting work in the field, and reveals novel results about the mechanisms of motor protein transport that will be broadly interesting. The assays are generally well performed, and the results and discussion are well described, but some descriptions in the article should be more rigorous and objective. The article is very long, and I think it would benefit from streamlining and reducing the number of figures to make it more accessible for non-specialists in the field.

Here are some concerns:

Major:

1. Fig. 1A shows the domain organization of all kinesin-3 members, but Figure 1B only represents KIF1B β , KIF13B and KIF16B as controls. Generally, the KIF1B α has the highest sequence similarity with KIF1C in kinesin-3 family (very high sequence similarity before aa 992 in KIF1C, which locates in IDR, probably contains IDR2a from Fig. S10A). In addition, both KIF1C and KIF1B α contain a PLD from the prediction in this paper (Figure S2C). Although the authors show the phenotype of KIF1B α in the Fig. S9, it might be better to put some descriptions up front, as readers may consider why the authors did not use KIF1B α as a control. Actually, I kept thinking about this concern before I got to the discussion.

Response: We thank the reviewer for this suggestion. We have moved the descriptions of KIF1B α phenotypes to earlier in the manuscript. We show that KIF1B α forms puncta in cells but unlike KIF1C, the KIF1B α puncta do not colocalize with known RNA granules P-bodies or stress granules (now in Fig S5 B,C). We show that, unlike KIF1C, the KIF1B α puncta do not incorporate GU-rich or polyA RNA (now in Fig S6 B).

2. It would be better if the authors can combine the Fig. 2B and 2C, since the article did not mention Fig. 2B at all. In addition, Fig. S3 does not help this article too much. Probably it would be better if the authors could take the Δ IDR-mNG data from the Fig. S3 and put into the Fig. 2. as a negative control, especially for Fig. 2D and 2F. As for whether the phenotype of the Δ IDR-mNG construct is "similar to a constitutively active KIF1C construct containing only the motor domain (amino acids 1-348) (Fig. S3 C)", I do not think it is important here, since in this part, the authors are aiming to confirm the IDR is critical for KIF1C phase separation.

Response: We have combined Figures 2B and 2C as suggested. We prefer to leave Figure S3 intact since, as the reviewer mentioned, the article is already long and these data are not critical for the story.

3. The description "the condensate properties can be modulated by adjacent coiled-coil segments" in the abstract and the sentence "However, the coiled-coil segments in the stalk domain appear to facilitate puncta formation as the addition of increasing amounts of coiled coil resulted in increased KIF1C enrichment in puncta as compared to the IDR alone" in the article are not accurate, since there is no direct evidence in this manuscript that shows that. In Fig. 2D, as well as Fig. 6A and Fig. S7, it is manifest at a glance there are lots of IDR-mNG localized in nucleus, which decreases the concentration of this construct in cytoplasm which in turn may lower its capability to form puncta. This is important, as the

results in Fig.4 show that the concentration of protein directly affects the formation of phase separated puncta in cells. From my view, the words "modulate", "tune" ... usually describe active processes, and these words may be confusing unless there are enough evidence support direct regulation. But the data presented in this article suggests to us that it is likely a passive process, such as the coiled coil region preventing the CC4-IDR construct from entering the nucleus (Fig. 2D, Fig. 6A and Fig. S7). Moreover, CC4 does affect the critical concentration of IDR in vitro (Fig. 5E), but that could be attributed to the coiled coil domain increasing its solubility. I like the word "influence" used in a subtitle in the discussion portion.

Response: We have removed this from the text.

In addition, the in vitro study of this paper in Fig. 5 did not show any significant difference of the puncta formation between IDR-mNG and CC4 - IDR-mNG (Diameter: $0.43 \pm 0.22 \mu\text{m}$ (mean \pm STD) for IDR-mNG vs $0.48 \pm 0.27 \mu\text{m}$ (mean \pm STD) for CC4-IDR-mNG. Roundness: No value was show in the article). So, a stricter assay or a more accurate description is required here to avoid any misleading to the readers.

Response: We now include p values showing that the differences in diameter and roundness are statistically significant (data moved to Fig S8 B).

The description for Fig. 5 "At 2 μM protein concentration and 100 mM NaCl, the KIF1C(IDR) droplets were smaller [diameter $0.43 \pm 0.22 \mu\text{m}$ (mean \pm STD)] than KIF1C(CC4+IDR) droplets [$0.48 \pm 0.27 \mu\text{m}$ (mean \pm STD)] (Fig. 5 C)" does not appear accurate as well, since there is no significant difference between the value $0.43 \pm 0.22 \mu\text{m}$ and the value $0.48 \pm 0.27 \mu\text{m}$, so it should not be described as "smaller". In addition, the article mentioned that "The KIF1C(IDR) puncta were also less round than those of KIF1C(CC4+IDR) (Fig. 5 C)", but there is no corresponding value from the quantification show the KIF1C(IDR) is less round.

Response: We now include p values showing that the differences in diameter and roundness are statistically significant (data moved to Fig S8 B).

4. The description in sentence "We thus tested whether ... LLPS is mutually exclusive (Fig. S5 A)" may not be accurate. Results in Fig. S5 only show there is no direct interaction between KIF1C and CLIP-170 or these two proteins do not colocalize. The words "mutually exclusive" means two proteins competent each other in the same location from my understanding.

Response: We have replaced the words "mutually exclusive" with "no colocalization" (line 204).

In addition, is it necessary to put Fig. S5 into this article? Since from my side, it does not help too much for the whole story. In cells, kinesin motors are autoinhibited in the cytoplasm. For this KIF1C, most of motors appear autoinhibited as well, even when the authors removed the IDR based on Fig. S3C (ΔIDR -mNG vs. MD-mNG). In this case, it is hard to investigate the potential interaction between the KIF1C (or its ΔIDR mutant) with the microtubules or with the tubulin due to the autoinhibition of constructs used in Fig. S5. It would be better to use other active versions of KIF1C, such as ΔP (Soppina et. al., PNAS, 2014) or other mutants (Ren et. al., PNAS, 2018; Wang et. al., Nat. Commun., 2022) if the authors want to show this part in the article.

Response: We agree that this data is not essential for the story, however, it may be of interest and benefit to others in the field studying LLPS of microtubule-associated proteins and we prefer to leave Figure S5 (now Figure S4) in the supplementary information.

5. The conclusion "This result suggests that the IDR- driven LLPS of KIF1C does not depend on mRNA incorporation, but is strongly affected by it" may not be accurate, there is no direct evidence that shows that mRNA, at least Rab13mRNA incorporation strongly affects the IDR- driven LLPS of KIF1C. Perhaps a knock out of Rab13mRNA would alter the formation of condensates, which would support a direct effect on LLPS.

Response: We have changed the text (line 306).

In addition, the sentence "These results also show that the LLPS is resistant to truncations of large portions of IDR" may not be accurate, from my view, except IDR2a, the rest of the IDR may not participate or contribute too much to the formation of puncta, but that doesn't mean LLPS is resistant to the truncation of these portions in IDR, these are different logics. The quantification from Fig. 7E also show there is no significant difference between the ST and truncations except Δ IDR2 and Δ IDR2a in statistics, such as ST (21.8 {plus minus} 12.0 puncta per cell, 2.06 {plus minus} 0.83 μ m diameter), Δ PLD (20.1 {plus minus} 13.7 puncta per cell, 1.62 {plus minus} 0.94 μ m diameter), Δ IDR1 (23.1 {plus minus} 14.3 puncta per cell, 2.13 {plus minus} 1.04 μ m diameter), Δ IDR3 (18.4 {plus minus} 8.1 puncta per cell, 1.71 {plus minus} 0.98 μ m diameter).

Response: We have changed the text (line 307).

6. I am not sure I agree with the author's interpretation of their FRAP data in Fig. 3. It appears to me that there is a large immobile population of molecules, as the bleached areas recover less than 50% of their initial intensity. However, the authors conclude that there is rapid exchange of molecules in the puncta. The authors need to further analyze and discuss both the exchange rate of the population of molecules that exchange, but also the fraction of apparently immobile molecules that do not recover in their experiments. These data appear to suggest that a large percentage of the molecules in the KIF1C puncta in fact do not exchange with the cytoplasm and undermine their argument for a liquid-like phase of the puncta.

Response: We agree that the data suggest the existence of an immobile pool of KIF1C within the condensates. We have added this information to the main text (lines 178-182). We note that these findings are consistent with recent studies demonstrating membrane-less organelles with at least partially solid-like properties, including nucleoli and stress granules as well as microtubule associated proteins (see references, reviewed in Van Treeck & Parker 2019).

Minor:

1. As mentioned above, Fig. 2 F needs a negative control, since the values of FL and IDR are lower than other constructs, maybe use the Δ IDR-mNG protein is better. In addition, from my view, the lower value of IDR construct does not represent this construct has lower capability to form puncta, but more likely because most of this protein localizes in nucleus, thus dramatically lowering the cytoplasmic concentration.

Response: We have changed the text as suggested (lines 152-154).

2. Fig. 6A probably need a negative control as well, maybe use the same construct Δ IDR in Fig. S7 is better.

Response: We have now included KIF1B α as a negative control (Fig S6 B).

3. Although I guess the reason for using hTERT-RPE1 cells in Rab13mRNA rescue assay (Fig. 6D-G) probably is easier to get KIF1C knock out cells (if I am correct), it would be better if there is a brief introduction for the reason to use hTERT-RPE1 here, since all previous assay in the article used COS-7 cells.

Response: You are correct and we have added text introducing the use of hTERT-RPE1 cells (line 269).

4. Is there any specific reason to use the construct ST in Fig. 7? Since in Fig. 6, the authors used FL-length KIF1C, if the authors want to avoid any effects caused by motor domain, the construct CC4-IRD also could be a simpler candidate.

Response: No specific reason other than to be consistent as most experiments that we carried out in cells used the ST construct (e.g. FRAP assay in Fig 3, hypotonic assay in Fig 3, RNaseA injection in Fig 4, RNA incorporation in Fig 4). (Note that Fig 7 is now Fig 6).

5. This article is a great case for motor-cargo interaction, since the RNA binding site of KIF1C is within its tail domain. This left me curious about if the interaction between the KIF1C and the membrane-less RNA granule is sufficient to release the KIF1C motor from autoinhibition? I guess the binding of RNA is not enough to release the KIF1C from autoinhibition. From Fig. S3C and Fig. 6D, seems the motor still in autoinhibition, even remove the Rab13mRNA binding region.

Response: We believe the question of whether the RNA binding relieves autoinhibition of KIF1C is beyond the scope of this manuscript and we plan to address this in the future with recombinant full-length KIF1C and *RAB13* mRNAs.

6. There are some grammar mistakes, e.g., There should be a "is" between "IDR" and "critical" in the title "A subregion of the KIF1C IDR critical for enrichment of Rab13mRNA in condensates".

Response: Thank you. We have corrected this (line 289).

7. There should be a definition for the full names of the abbreviate "RBD" mentioned in the article although the readers may guess that is an RNA binding domain, if possible, it would be better but not necessary if the authors could show the residues or the region in IDR.

Response: RBD is defined at the beginning to the section "KIF1C condensates display properties of RNA granules" (line 219) but in response to the reviewer's comment, we now include this definition a second time in the Discussion section (line 420).

8. In the results (line 126), the authors refer to the KIF1C IDR without first defining this region in the introduction. I would re-word this sentence for clarity by first defining what an IDR is and how it's assessed in the current study.

Response: The IDR is defined at the end of the Introduction (lines 94-95).

9. What is the significance of the roundness measurement in Fig. 5? This should be described for the reader.

Response: Roundness refers to the shape of the droplet and this is now included in the text (line 323, data moved to Fig S8 B).

10. The authors state several times that this is the first kinesin shown to undergo LLPS. However, is this true? What about the recent work showing that the yeast Tea2 kinesin undergoes LLPS with other +TIP components (Maan et al. NCB 2023).

Response: We thank the reviewer for this comment. The recent work from the Dogterom lab (Maan et al., 2023) demonstrates that the end binding (EB) protein Mal3 forms condensates alone and with the kinesin-7 family member Tea2 and its cargo Tip1 for enrichment at microtubule plus ends. The authors show images of Mal 3 droplets and the requirement of the IDR domain and the crowding agent polyethylene glycol for droplet formation. The authors state that "*Tea2 and Tip1 formed condensates under similar crowding conditions and concentrations on their own (Extended Data Fig. 5).*" However, Extended Data Fig 5 reports on the fluorescence intensity of Mal3-EGFP colocalizing with Tea2 or Tip1. No images of Tea2-only droplets are shown and no information is provided on the Tea2 and/or PEG concentrations required for droplet formation or the liquid nature of Tea2 droplets. Thus, we do not feel comfortable stating that Tea2 on its own undergoes LLPS.

We do reference the Maan et al., 2023 work in the Discussion listing microtubule-associated proteins shown to undergo LLPS (line 403) and when comparing the μM concentrations of KIF1C required for LLPS to the mM concentrations of these other microtubule-associated proteins (line 417).

11. The authors don't discuss KIF5A, but their analysis reveals it also contains a low complexity region that may undergo LLPS (Fig. S2D). This would fit with recent reports that KIF5A tends to oligomerize more than other KIF5 isoforms, and that mutations in KIF5A that impact the tail domain may lead to aberrant oligomerization. I feel that it would be useful to the field for the authors to discuss these results in light of their own.

Response: We thank the reviewer for this suggestion. Although it is intriguing that KIF5A is predicted to contain an IDR, there is, however, no data to suggest that KIF5A undergoes LLPS. Rather, the current literature suggests that KIF5A undergoes higher-order oligomerization and accumulation at the cell periphery, especially for the isoform lacking exon 27 (Nakano et al., 2022, Baron et al., 2022, Pant et al., 2023, Soustelle et al., 2023). It thus does not seem prudent for us to speculate on whether or not KIF5A undergoes LLPS.

Reviewer #3 (Significance (Required)):

The study is novel and interesting and will be impactful for the cytoskeletal and RNA biology communities. The experiments are of high quality and controls are appropriate. The finding that motor proteins can participate in LLPS will be of high interest for a variety of fields and provides a very interesting advance over current knowledge in the field.

Dear Dr Verhey,

Thank you submitting a revised version of your manuscript to the EMBO Journal via Review Commons. It was sent to the same three reviewers that originally appraised your work; I have now received reports on this revised version from two of them. Their comments are attached to the bottom of this email. As you will see, both referees are satisfied with the changes you made. I would, however, like you to incorporate the concerns of Referee #1 into the discussion section of the manuscript. Before we can move forwards towards publication of your manuscript, there are also some remaining editorial points which need to be addressed. In this regard, would you please:

indicate the corresponding author in the manuscript file,
include five keywords,
include a 'Disclosure and competing interests statement',
complete the author checklist form,
upload each figure as separate high-resolution files,
include each appendix file in PDF format,
move Figures S1-S8 to a new file Appendix PDF starting with a table of contents with page numbers,
rename figures S1-S8 'Appendix Figure S1-S8' with the callouts changed in the text as appropriate,
place appendix figure legends below the corresponding figures in the Appendix PDF,
indicate the statistical test used for data analysis in the legend of supplementary figure 6c,
define the scale bar in the legend of supplementary figures 7c-d,
define the pink and white arrows in the legend of supplementary figures 7c, e, and the orange, green and pink arrows in the legend of figure 1d.
rename movie files as Movie EV1-EV10 with the corresponding text callouts, zipping the corresponding legend with each,
correct the section order as follows: title page with complete author information, abstract, keywords, introduction, results, discussion, materials & methods, data availability section, acknowledgements, disclosure and competing interests statement, references, main figure legends, tables, expanded figure legends.

We include a synopsis of the paper on our website (see <http://emboj.embopress.org/>). Please provide me with a general summary image, two-sentence summary statement and 3-5 bullet points that capture the key findings of the paper.

I look forward to receiving these changes. EMBO Press is an editorially independent publishing platform for the development of EMBO scientific publications.

Best wishes,

William

William Teale, PhD
Editor
The EMBO Journal
w.teale@embojournal.org

- a point-by-point response to the referees' comments, with a detailed description of the changes made (as a word file).
 - a word file of the manuscript text.
 - individual production quality figure files (one file per figure)
 - a complete author checklist, which you can download from our author guidelines (<https://www.embopress.org/page/journal/14602075/authorguide>).
 - Expanded View files (replacing Supplementary Information)
- Please see out instructions to authors

We realize that it is difficult to revise to a specific deadline. In the interest of protecting the conceptual advance provided by the work, we recommend a revision within 3 months (21st Jul 2024). Please discuss the revision progress ahead of this time with the editor if you require more time to complete the revisions. Use the link below to submit your revision:

Referee #1:

The authors have done a good job responding to my concerns and I believe the paper should be published in the EMBO Journal. One point that I would still make, is that I believe the authors are still overstating the novelty of their findings by suggesting this is the "first" demonstration of LLPS for a kinesin molecule (in at least 3 different places in the manuscript). The authors cite Maan et al. but claim in the rebuttal letter that this paper does not demonstrate a kinesin molecule undergoing LLPS on its own. This may be true, but that paper does show an orthogonal kinesin molecule (Tea2) that enters LLPS droplets in cells and in vitro, and also demonstrates functional activity of the kinesin within the phase separated MT tip complex in vitro through motility assays. I think the authors are splitting hairs here to claim that Maan et al. does not show evidence of LLPS for Tea2 alone, and their multiple claims of primacy in the current manuscript should be toned down in the final version of the manuscript.

Referee #2:

In the revised manuscript the authors have made a good effort to address my concerns. The antibody staining in WT and KO cells provides good evidence that Rab13 mRNA/KIF1C positive structures are found at the endogenous expression level at the cell periphery. This is further supported by imaging cells expressing low levels of labelled protein with and without the IDR, as requested.

One could question whether these structures are indeed condensates (The main text subheading and Figure 8 titles perhaps overstate this), but taking the manuscript as a whole, a strong case is made for new and exciting concept for the field, that merits publication in a leading journal.

Note: I think the magenta panels in Figure 8F are mislabeled - should this be Rab13 mRNA?

EMBOJ-2024-117339-T Response to reviewer's comments.

Referee #1:

The authors have done a good job responding to my concerns and I believe the paper should be published in the EMBO Journal. One point that I would still make, is that I believe the authors are still overstating the novelty of their findings by suggesting this is the "first" demonstration of LLPS for a kinesin molecule (in at least 3 different places in the manuscript). The authors cite Maan et al. but claim in the rebuttal letter that this paper does not demonstrate a kinesin molecule undergoing LLPS on its own. This may be true, but that paper does show an orthogonal kinesin molecule (Tea2) that enters LLPS droplets in cells and in vitro, and also demonstrates functional activity of the kinesin within the phase separated MT tip complex in vitro through motility assays. I think the authors are splitting hairs here to claim that Maan et al. does not show evidence of LLPS for Tea2 alone, and their multiple claims of primacy in the current manuscript should be toned down in the final version of the manuscript.

Response: We have altered the text to tone down our claims.

Referee #2:

In the revised manuscript the authors have made a good effort to address my concerns. The antibody staining in WT and KO cells provides good evidence that Rab13 mRNA/KIF1C positive structures are found at the endogenous expression level at the cell periphery. This is further supported by imaging cells expressing low levels of labelled protein with and without the IDR, as requested.

One could question whether these structures are indeed condensates (The main text subheading and Figure 8 titles perhaps overstate this), but taking the manuscript as a whole, a strong case is made for new and exciting concept for the field, that merits publication in a leading journal.

Note: I think the magenta panels in Figure 8F are mislabeled - should this be Rab13 mRNA?

Response: We apologize for the confusion. The figure legend was mistaken and has been corrected. The magenta panels indicate anti-KIF1C immunostaining. In addition, we reorganized the panels to make it less confusing (panel F is now panel D). In the revised version, panels A-D all focus on validation of the antibody and localization of KIF1C whereas panels E and F focus on RNA localization.

Dear Kristen,

I am pleased to inform you that your manuscript has been accepted for publication in the EMBO Journal.

Congratulations on a really lovely study!

Best wishes,

William

William Teale, PhD
Editor
The EMBO Journal
w.teale@embojournal.org
